# Microbiomes in the Challenger Deep slope and bottom-axis sediments

Ying-Li Zhou [1,2,5], Paraskevi Mara [3,5], Guo-Jie Cui[1,2], Virginia P. Edgcomb [3] & Yong Wang [1,4 ✉]

Hadal trenches are the deepest and most remote regions of the ocean. The 11-kilometer deep Challenger Deep is the least explored due to the technical challenges of sampling hadal depths. It receives organic matter and heavy metals from the overlying water column that accumulate differently across its V-shaped topography. Here, we collected sediments across the slope and bottom-axis of the Challenger Deep that enable insights into its in situ microbial communities. Analyses of 586 metagenome-assembled genomes retrieved from 37 metagenomes show distinct diversity and metabolic capacities between bottom-axis and slope sites. 26% of prokaryotic 16S rDNA reads in metagenomes were novel, with novelty increasing with water and sediment depths. These predominantly heterotrophic microbes can recycle macromolecules and utilize simple and complex hydrocarbons as carbon sources. Metagenome and metatranscriptome data support reduction and biotransformation of arsenate for energy gain in sediments that present a two-fold greater accumulation of arsenic compared to non-hadal sites. Complete pathways for anaerobic ammonia oxidation are predominantly identified in genomes recovered from bottom-axis sediments compared to slope sites. Our results expand knowledge of microbially-mediated elemental cycling in hadal sediments, and reveal differences in distribution of processes involved in nitrogen loss across the trench.

[1] Institute of Deep-Sea Science and Engineering, Chinese Academy of Sciences, Sanya, Hainan, China. [2] University of Chinese Academy of Sciences, Beijing, China. [3] Department of Geology and Geophysics, Woods Hole Oceanographic Institution, Woods Hole, MA, USA. [4] Present address: Institute for Ocean Engineering, Shenzhen International Graduate School, Tsinghua University, Shenzhen, China. [5] These authors contributed equally: Ying-Li Zhou, Paraskevi Mara. ✉email: wangyong@sz.tsinghua.edu.cn

Hadal trenches are geological formations isolated in deep-sea environments (6000–11,000 m depth) that account for the deepest 45% of the oceanic depth range[1,2]. Technological challenges constrain sampling and bathymetric observations in hadal trenches, placing them among the least-explored marine environments.

The Challenger Deep (CD) is the deepest part of the world's oceans located in a tectonically active area at the southern end of the Mariana Trench in the Western Pacific. CD experiences frequent shallow earthquakes[3] that are likely to trigger gravity flows of sedimentary and volcanic material from the adjacent margin into the hadal zone[4,5]. The V-shaped topography of the trench enables the collection of significant amounts of particulate organic matter (POM) from the upper water column and abyssal seafloor to CD. The average particulate organic carbon accumulation rates in southern CD seafloor are estimated roughly at $\sim 1.5 \times 10^{-5}$ g cm$^{-2}$ yr$^{-1}$ [6]. Previous studies demonstrated that bottom-axis sediments (10,817 m depth), have higher organic carbon content[7], and more intense organic matter diagenesis than adjacent slope sites[7,8]. Degradation of sedimentary organic matter and detrital proteins in the trench is expected to sustain hadal microbial activity[5,7] and to provide substrates like ammonia that can be used by autotrophic ammonia-oxidizing archaea[5,7,9,10] to contribute to global carbon cycle. Functional marker gene studies from other hadal realms (e.g., Ogasawara Trench; 9760 m) revealed anaerobic ammonia oxidizers (anammoxers)[11], suggesting that nitrogen (N$_2$) production occurs in hadal sediments via ammonia and nitrite utilization. However, detailed genomic analysis is required to provide a complete overview of N$_2$ cycling in the hadal zone.

Volcanic ash from Earth's upper crust is enriched with heavy metals such as mercury, arsenic, and selenium[12], and can lead to the accumulation of heavy metals in deep-sea locations such as CD. Sinking POM can also deliver heavy metals to hadal sediments[13,14] and the funneling effect in topographically isolated trenches like CD may further enhance their accumulation. Elemental analysis of toxic metals confirmed bioaccumulation of arsenic in hadal fish collected from the Mariana Trench at concentrations ~30 p.p.b[15]. If concentrations do not reach toxic levels, we propose that microbial inhabitants of CD sediments may be able to use heavy metals (e.g., arsenate-As(V) and arsenite-As(III)) as alternative electron acceptors for energy gain as was proposed for an early Earth anoxic ocean[16]. Microbial biotransformations can mobilize heavy metals into less or non-toxic analogs that can confer osmoregulation advantages and protection to marine bacteria in low- or high-temperature extremes[17,18]. The unique environmental conditions of CD coupled with its topographical isolation may drive microbial endemism, niche specialization, and the development of novel strategies for surviving hostile and energy-limiting conditions as described elsewhere[19,20]. 16S rRNA gene studies describing the prokaryotic communities in CD sediments are limited[21,22], and the metabolic activities of microbial communities across the different CD topographies have only been described on the basis of the SSU rRNA gene clone libraries, and quantitative polymerase chain reaction (PCR) analyses of functional nitrogen cycling-related genes from four bottom-axis sediment samples at ~10 km depth[9].

Here we present 586 prokaryotic metagenome-assembled genomes (MAGs) from 37 sediment metagenomes covering north and south slope sites and bottom-axis sites of CD. Our data show that the overall microbial community composition is different between slope and bottom-axis sites. Anoxia and nutrient availability shape the community structure in CD and are likely strong selective forces on the microbial diversity detected along geochemical gradients in CD sediments. In situ microorganisms encode complete denitrification and anammox pathways in their genomes; however, the distribution of anammoxers and denitrifiers appears heterogeneous along the trench and with sediment depth. Active transcription of genes involved in biotransformations of arsenic and selenium suggests the presence of ancient metabolisms used for energy gain. Genomic and metatranscriptome data for bottom-axis and slope sites are used to determine whether the inferred metabolic capacity and ecological roles of microorganisms found across the slope and bottom-axis sediments show distinctions and whether they differ from what is observed in studies of abyssal and other hadal deep-sea sediments in the region[23,24].

## Results and discussion

**Geochemistry of CD sediments.** We collected CD sediment cores from 11 sites (5183–10,911 m depth; Supplementary Data 1 and Supplementary Fig. 1a) for porewater nutrients analyses. Distinct nutrient conditions existing in slope and bottom-axis sediments have been previously described[7,8,22] in CD and were also detected in our study. Relatively high NO$_3^-$ (14–35 μM) and O$_2$ (134–152 μM) concentrations were observed in all surface layers. Sharp decreases in NO$_3^-$ (down to 0.6 μM) and O$_2$ (down to 1 μM) concentrations at 18–35 cm below seafloor (cmbsf) were detected only at the bottom-axis sites, while NO$_3^-$ and O$_2$ concentrations had negligible changes with depth at the different slope sites examined (Supplementary Fig. 1b). NH$_4^+$ concentrations presented a sharp increase with depth in the bottom-axis sediments reaching up to 18.3 μM at depths below 20 cmbsf (see Supplementary Discussion). The measured temperature at our deepest site was ~2.5 °C. We also estimated the total arsenic (As) concentrations in 13 CD sites, and in 6 non-hadal sites (Supplementary Fig. 1a). As concentrations ranged between 2.2 and 11 μg/g of dry sediment. Bottom-axis sediments presented higher As concentration (26% on average) compared to slope sites (Supplementary Fig. 1c), while both slope and bottom-axis sediments had >2-fold higher total As compared to non-hadal reference sites (Wilcoxon test, $p < 0.001$). We detected similar trends for total selenium and mercury concentrations that were also measured in our CD sediments (Supplementary Fig. 1d, e). As(V) accounted for ~25% (on average) of the measured total As and showed accumulation with depth (Supplementary Data 1). The rest of the ~75% of As pool was most likely accounted for by organoarsenical species, although these organoarsenicals were not specifically identified using Atomic Fluorescence Spectrometry. As(III), monomethylarsonic acid (MMA), and dimethylarsinic acid (DMA) were below the detection limit (<0.2 μg/g) (see "Methods").

**Microbes in CD sediments.** Samples from different depth horizons of recovered sediment cores from 13 sites were used to generate 37 metagenomes (Fig. 1a, Supplementary Fig. 2, and Supplementary Data 2). 16S rDNA fragments (16S miTags) were extracted from clean reads of these metagenomes to describe the microbial communities of CD[25]. More than 80% of the 16S miTags in our samples were affiliated with bacteria, while 17% of them were assigned to archaea (Supplementary Fig. 3). The remaining 3% of the 16S miTags were unclassified prokaryotes. The most abundant phyla in the CD sediments were Proteobacteria (Alphaproteobacteria) and Thaumarchaeota (Nitrososphaeria) (Supplementary Fig. 3).

The percentages of the novel (defined as <97% identity to references in SILVA 138 SSU database) 16S miTags that were recovered per metagenome from CD sediments (26 ± 7%; $n = 37$) were significantly higher compared to those collected from bathyal or abyssal sediments from the South China Sea, Mariana

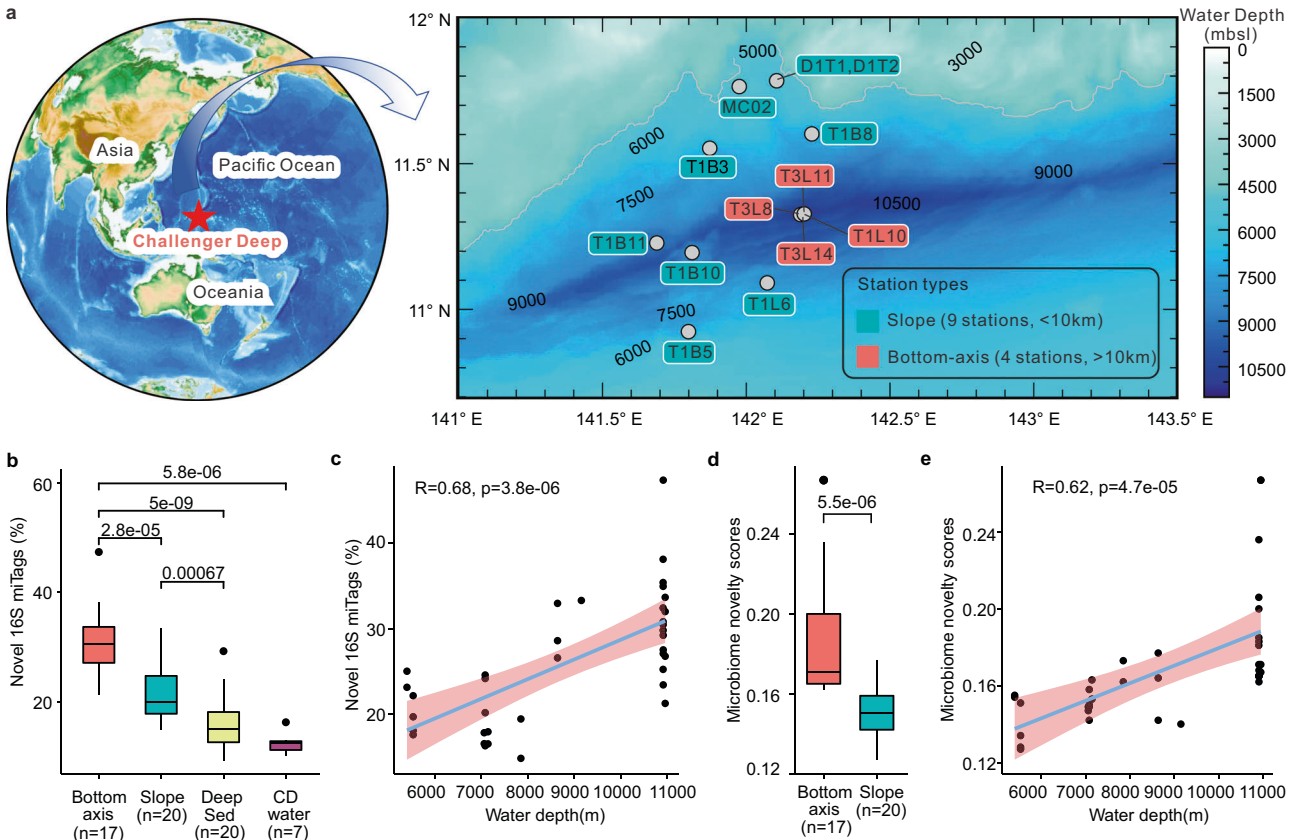

**Fig. 1 Overview of sampling sites and microbiome novelty in Challenger Deep (CD) sediments. a** Sampling locations in CD (details of the sampling sites are in Supplementary Data 2). **b** Box plots showing the average percentage of novel 16S miTags in slope and bottom-axis CD sediment metagenomes (this study), deep-sea hydrothermal/cold-seep sediment metagenomes (Deep Sed), and CD hadal water column (CD water) metagenomes from public databases ($p$ values were estimated using the two-sided Wilcoxon test for pairwise comparisons). Sequences and identity results are deposited in https://doi.org/10.6084/m9.figshare.12979709. **c** Pearson's correlation analyses between the estimated novel 16S miTags and the different depths of the sampling sites ($p$ value was calculated using the two-sided $t$ test). **d** Microbiome novelty scores (MNS) for CD slope and bottom-axis sediments ($p$ values estimated using the two-sided Wilcoxon test). **e** Pearson's correlation analyses between the microbiome novelty scores and the different depths of the sampling sites ($p$ values were calculated using the two-sided $t$ test). In box plots (**b**, **d**), center lines indicate median values. The lower and upper bounds represent 25th and 75th percentiles, respectively. The lower/upper whiskers represent minima/maxima no further than 1.5 times the interquartile range from the hinge, and the points falling outside of the whiskers represent the outliers. The pink background at **c**, **e** indicates 95% confidence interval. Source data are provided as a Source data file.

Trench (NCBI BioProject accession PRJNA784430), deep-sea hydrothermal (Mid-Atlantic Ridge[26] and Guaymas Basin[27]) and cold seep (Gulf of Mexico[28,29]) sediment metagenomes ($16 \pm 5\%$; $n = 20$), as well as CD ($13 \pm 1.8\%$; $n = 7$) water column[30] (Wilcoxon test, $p < 0.05$) (Fig. 1b). The percentages of novel 16S miTags in the bottom-axis sediments were also significantly higher than those from the slope sites (Fig. 1b; Wilcoxon test, $p < 0.001$), and increased with both water and sediment depth (Fig. 1c and Supplementary Fig. 4a). Anaerobic microbial communities such as those detected in bottom-axis sediments are underrepresented in existing databases. Supplementary Fig. 4 indicates that increasing novelty scores in bottom-axis sediments are primarily driven by sediment depth and anoxia. Nutrient availability, which is often lower with depth, and anoxia are selective forces known to structure microbial communities in deep-sea sediments[31].

We also estimated the microbiome novelty score (MNS)[32] for all CD sediment samples as a proxy for microbiome novelty (Fig. 1d). The MNS was calculated based on the compositional uniqueness of each CD sediment microbiome when searched against public databases that contain microbiomes from diverse marine sediments in the Microbiome Search Engine 2[33]. The

estimated MNS for all CD samples was >0.12, which is the cutoff for novel microbiomes[17] (Fig. 1d). The MNS was significantly higher for bottom-axis samples compared to slope sediment samples (Wilcoxon test, $p < 0.001$) (Fig. 1d) and was positively correlated with both water and sediment depth (Fig. 1e and Supplementary Fig. 4b).

Taxonomic sorting of 3095 18S rRNA eukaryotic miTags from three depths at 5400, 7143, and ~10,900 m showed a dominance of fungal signatures from the phylum Ascomycota (82%) and specifically the class Sordariomycetes (Supplementary Fig. 5). Sordariomycetes are organic matter decomposers that encode a substantial number of carbohydrate-active enzymes[34]; however, their role and functions in the deep sea are still unknown. The dominance of Sordariomycetes and the presence of other saprophytic fungal taxa in our data indicate that fungi are potential recyclers of organic matter and nutrients in CD that could provide energy and labile carbon to hadal microbial communities, as described in other extreme environments[35]. However, the dominance of fungal signatures within eukaryotic metagenome data needs to be interpreted with caution. This is because the larger genomes of many eukaryotes will not be assembled into high-quality MAGs without deeper sequencing effort.

**Reconstruction and characterization of CD genomes**. Assembly of the 37 metagenomes generated 874,671 scaffolds (≥2000 bp, Supplementary Data 3) for genome binning of 586 good-quality (>50% completeness and <10% contamination) draft genomes, among which 125 were high-quality MAGs (>90% completeness and <5% contamination) (Fig. 2a and Supplementary Data 4). Quality controlled reads from the 37 metagenomes had high mapping rates to all MAGs (average 54%, Supplementary Data 3), suggesting a good representation of the MAGs in recovered microbiomes. Classification of the MAGs using the Genome Taxonomy Database[36] (GTDB) showed 33 archaeal and 553 bacterial MAGs representing 383 prokaryotic species (Supplementary Data 5) affiliated to 5 archaeal and 29 bacterial phyla (Fig. 2a).

**Spatial distribution of the microbes in CD**. To examine if distinct microbial communities exist between slope and bottom-axis CD sediments, we estimated the relative abundance and the prevalence (defined as the number of metagenomes in which >10% of one MAG was covered by clean reads[37]) of each of the 586 recovered MAGs. 154 and 262 MAGs were unique to bottom-axis and slope sediments, respectively, and 170 MAGs were shared between slope and bottom-axis samples (Supplementary Fig. 6). Principal coordinate analysis (PCoA) using the relative abundance of MAGs and the Bray–Curtis dissimilarity index confirmed a discrete separation of microbial communities between slope and bottom-axis sediments (Fig. 2b). This spatial separation was more distinct compared to what was observed previously using 16S rRNA data[21]. In our study, we also observed a community composition in CD sediments that is distinct in comparison to previously reported communities in the CD water column[29] (Supplementary Fig. 7).

While under-sampling of diversity is a possibility, we believe that many/most of the MAGs were identified solely from bottom-axis or slope CD sediments. The identification of MAGs unique to the different geographies (slope and bottom-axis) suggests microbial niche specificity (Supplementary Fig. 6). Unique habitats in CD can result from the accumulation of different nutrients or electron acceptors (e.g., $NH_4^+$, $NO_3^-$, $O_2$) in bottom-axis vs. slope sediments, and/or from the spatial isolation of the bottom-axis vs. slope sampling sites. In situ microbial populations may survive in these two different hadal locations by utilizing different suites of available nutrient sources. Our geochemistry data ($NH_4^+$ and $NO_3^-$ in slope vs. bottom-axis sites) support this hypothesis.

To further investigate the described spatial distribution of CD microbes we compared the average predicted microbial genome size from the 80 most prevalent MAGs for slope, bottom-axis, and ubiquitous CD-distributed (present in both slope and bottom-axis sediments) microbes. The average genome size of the ubiquitous CD-distributed microbes was ~16% larger when compared to the MAGs unique to the slope sites, and ~11% larger when compared to the unique bottom-axis MAGs (Fig. 2c). We suggest that reduced genome size could be an adaptation that lowers the metabolic cost associated with microbial DNA replication in CD sediments. Genome reduction can provide increased fitness under the nutrient limiting conditions[38] found in the bottom and slope hadal CD sediments. Bacterial genomes tend to increase in size by aggregating adaptive gene modules that can provide greater metabolic flexibility[39]. Metabolic flexibility might be an advantage for CD microbes inhabiting both slope and bottom-axis sites where different (and likely ephemeral) available energy pools exist. Indeed, we identified significantly more genes coding for carbohydrate-active enzymes (CAZymes)[40] and peptidases (t test, $p < 0.05$) in MAGs with ubiquitous distribution in CD (Supplementary Fig. 8), suggesting that microbes present in both slope and bottom-axis sediments encode a repertoire of enzymes that can support survival under different carbon and nutrient available conditions.

**Functional characterization of the CD genomes**. The potential ecological role of the microbial lineages in CD sediments was addressed by assigning metabolic functions to 1,469,633 predicted genes present in CD MAGs, and by examining the 3 metatranscriptomes (6–9, 12–15, 19–21 cmbsf of T3L11 core; 10,908 m depth) for actively transcribed metabolic genes. ~23% of the genes in CD MAGs had an unknown function and ~90% of these genes had no homologs in public databases. In addition, the three metatranscriptomes showed active transcription of 82,337 genes of unknown function (16% of the total transcribed genes) (Supplementary Data 4). On average, 9.1 and 4% of transcriptomic reads that mapped to MAGs were assigned to two species clusters of Gemmatimonadota and Marinisomatota in the bottom-axis site at 10,908 m (Supplementary Fig. 9). Functional genes detected in CD MAGs were used to reveal the potential lifestyle and adaptive strategies used by microorganisms in CD sediments. The large fraction of genes with unknown functions within the 37 metagenomes warrants future efforts to disclose their contribution to ecological adaptations and metabolic potentials.

**Recycling of detrital organic matter and utilization of hydrocarbons**. Due to the funneling effect, the V-shaped CD accumulates organic debris at different rates along the trench axis and with water depth[7,41]. Accumulation and degradation of organic matter and proteinaceous material are common in deep-sea/abyssal sediments[23,42–44] as well as in sediments with limited inorganic nitrogen sources[45]. We detected genes for 32,178 CAZymes and 34,791 peptidases in CD MAGs that could facilitate the breakdown of organic matter and proteinaceous compounds (Supplementary Figs. 10–12; Supplementary Data 4 and 6, and Supplementary Discussion). Enhanced carbohydrate and protein cycling in organic-rich trenches such as the Izu-Bonin Trench was posited to expand niche availability[44]. This can also be the case for CD hadal sediments considering the distinct nutrient pools that exist across the slope vs. bottom-axis sites, and the variety of CAZymes and peptidases encoded in CD hadal sediment MAGs (see also Supplementary Discussion). The necessity to recycle protein/organic matter might trigger microbial syntrophic interactions in CD sediments. Peptides released into these sediments can be further hydrolyzed to amino acids as described for other extreme environments[46]. Possible heterotrophic mechanisms in CD may include use of the conserved glycine cleavage system that breaks down glycine to C1 compounds and $NH_4^{+}$ [47]. The gcvT gene (gcv operon), which cleaves glycine to $CO_2$ and $NH_4^+$, was present in 475/586 of our MAGs and covered 30/34 identified phyla. The $CO_2$ and $NH_4^+$ produced by amino acid recycling (e.g., glycine cleavage system) could be a valuable carbon/nitrogen source for CD $CO_2$ fixers and anammoxers, respectively (Supplementary Fig. 13 and Supplementary Discussion).

Degradation of hydrocarbons by microbial sediment communities is known to occur in deep sediments[28,48]. CD receives saturated alkanes from sinking particles that accumulate in the sediments of the trench[49]. We report various genes involved in the aerobic and anaerobic degradation of n-alkanes and benzene rings (Fig. 3, Supplementary Data 7, and Supplementary Discussion), and a higher number of CD sediment MAGs with the metabolic potential to utilize hydrocarbons compared to what was observed in CD waters[49]. This suggests the potential use of hydrocarbon degradation for energy gain in surficial as well as

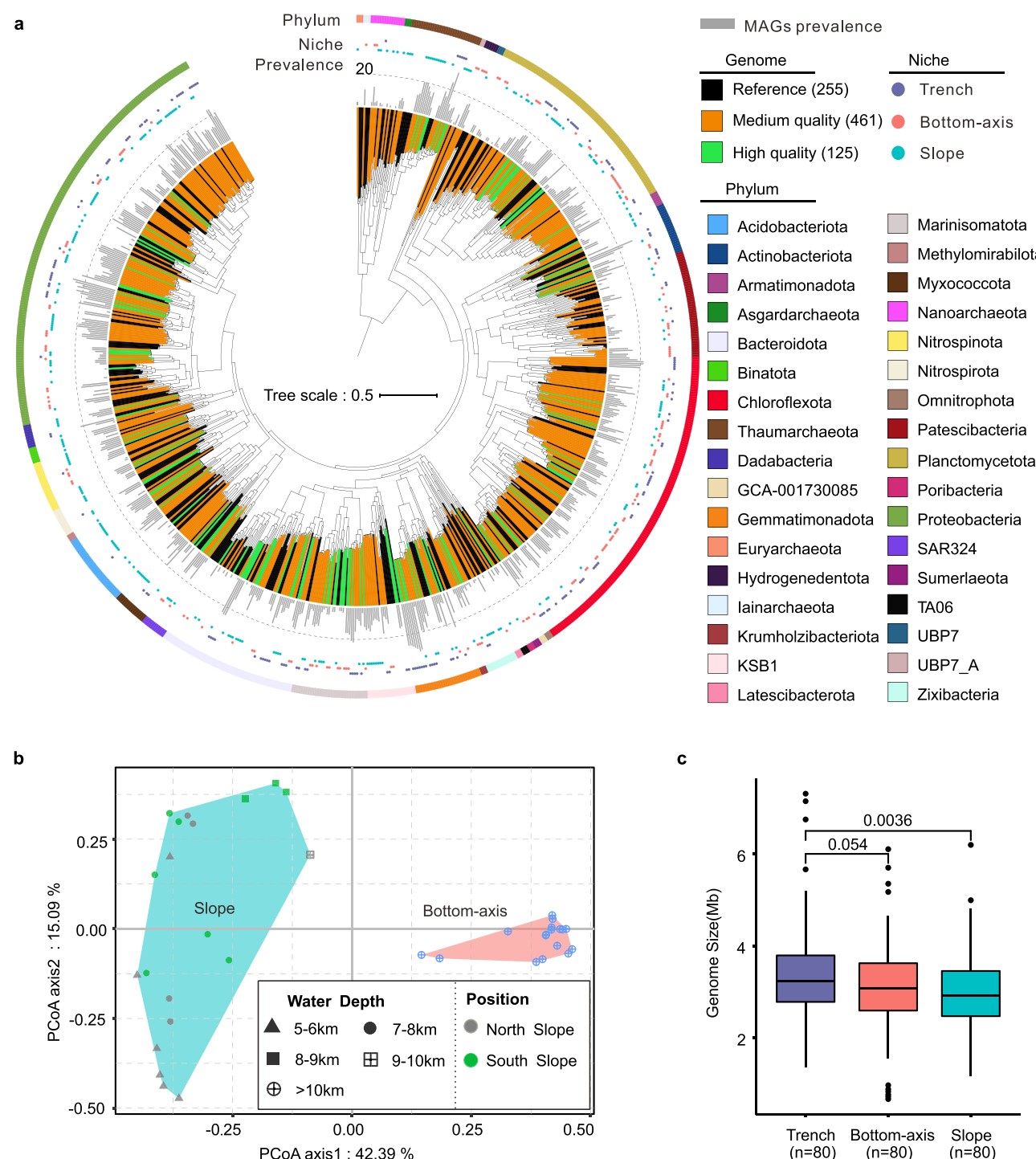

**Fig. 2 Phylogeny, taxonomy, and genome size of reconstructed metagenome-assembled genomes (MAGs). a** Maximum-likelihood phylogenomic tree of the 586 MAGs (green and orange clades present high and medium quality MAGs, respectively) and 255 reference genomes (black clades) from GTDB. The cyan dots present MAGs unique to slope sediments, the red shows MAGs unique to bottom-axis sediments, and the purple dots represent MAGs found in both slope and bottom-axis sediments (Trench). Gray bars show the prevalence of the MAGs in the 37 metagenomes. **b** PCoA ordination analysis using the relative abundance of MAGs and Bray-Curtis dissimilarity index. **c** Comparison of estimated genome size of the top 80 prevalent MAGs found in bottom-axis (Bottom), slope (Slope), and in both slope and bottom-axis sediments (Trench). *p* values were estimated with a two-sided *t* test analysis using the Trench MAGs as reference. In the box plot, center lines indicate median values. The lower and upper bounds represent 25th and 75th percentiles, respectively. The lower/upper whiskers represent minima/maxima no further than 1.5 times the interquartile range from the hinge, and the points falling outside of the whiskers represent the outliers. Source data are provided as a Source data file.

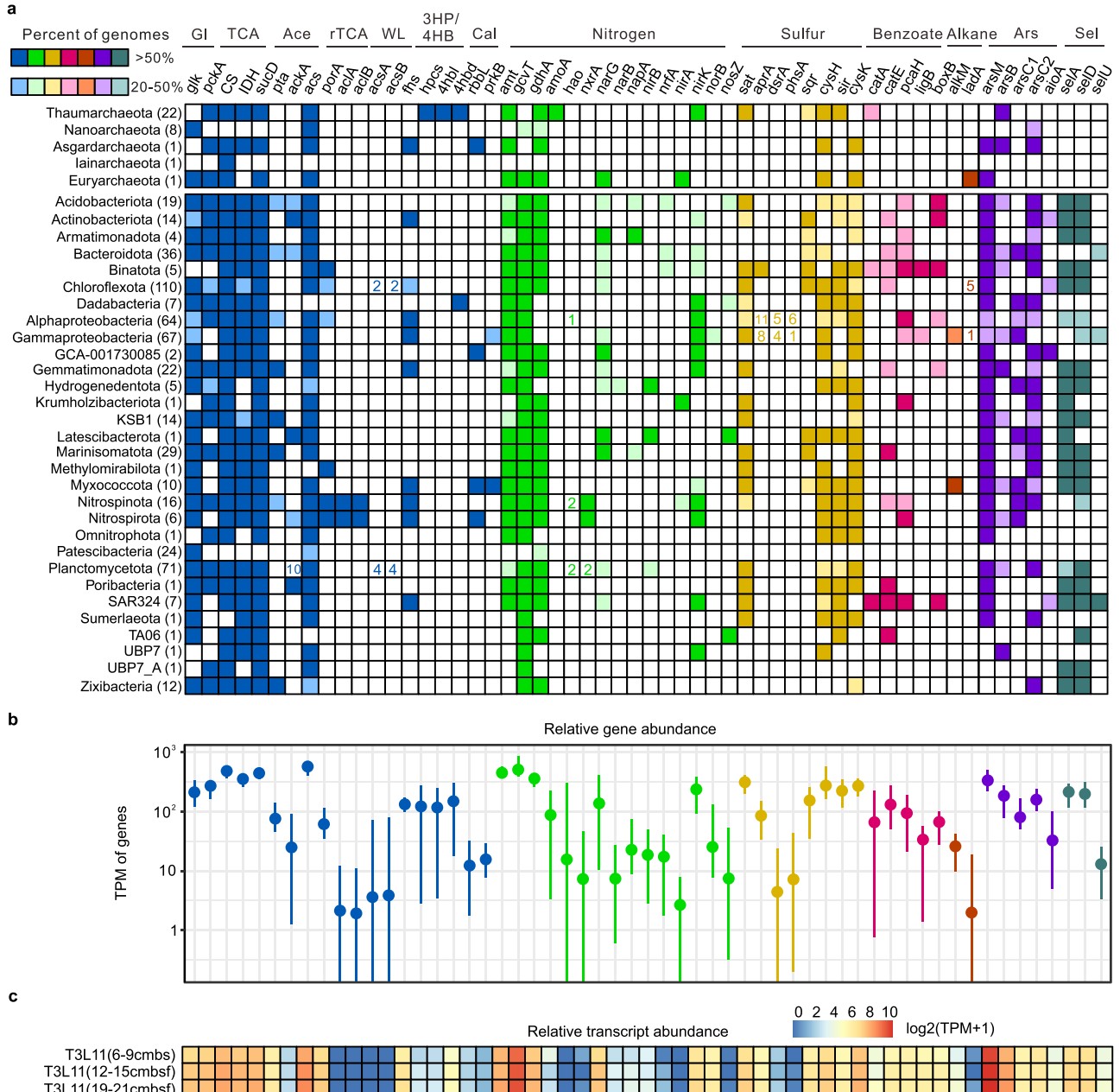

**Fig. 3 Presence of selected metabolic genes in the 37 metagenomes and the three bottom-axis sediment metatranscriptomes. a** Heatmap of various metabolic genes across phyla (class level for Proteobacteria) found in the CD sediments. Light colors indicate genes present at 20–50% in the retrieved MAGs. Darker colors indicate genes present at 50–100% in the retrieved MAGs. The number of genomes per phylum is shown in the parentheses. The number of key genes present in <20% of genomes is given in the squares. **b** Mean relative gene abundances across the 37 metagenomes. The bars indicate the minima and maxima of the gene abundances. **c** Heatmap of the relative abundance of transcripts (TPM) in the three metatranscriptomes (6–9, 12–15, 15–21 cmbsf) from the bottom-axis sediment sample at the T3L11 site (10,908 m). Gl glycolysis and gluconeogenesis, TCA tricarboxylic acid cycle, rTCA reductive tricarboxylic acid cycle, Ace acetate metabolism, WL Wood–Ljungdahl pathway, 3HP/4HB 3-hydroxypropionate/4-hydroxybutyrate cycle, Cal Calvin cycle, Asr arsenate metabolism, Sel selenate metabolism, TPM transcripts per million. Details of the metabolic genes can be found in Supplementary Data 7. Source data are provided as a Source data file.

deeper sediments (>10 cmbsf) where $O_2$ drops drastically, particularly at bottom-axis sites.

**$CO_2$ fixation.** Evidence for autotrophic $CO_2$ fixation that can provide labile organic carbon to CD microorganisms, was detected in our MAG data. Autotrophic processes have been reported to dominate in the CD water column[50]; however, in abyssal and hadal sediments obligate autotrophic taxa are reported either to be absent[23] or to perform specific autotrophic pathways, as observed in the Yap Trench[24]. Similar to what was found in the Yap Trench, we detected genes of the 3HP/4HB pathway in both slope and bottom-axis metagenomes (Supplementary Fig. 14). These genes were the most abundant among all $CO_2$ fixation pathways and were assigned to Thaumarchaeota (Fig. 3; see also Supplementary Discussion). 3HP/4HB genes were also highly transcribed in the bottom-axis metatranscriptomes (Fig. 3c). Key genes (e.g., ATP-citrate lyase) that participate in

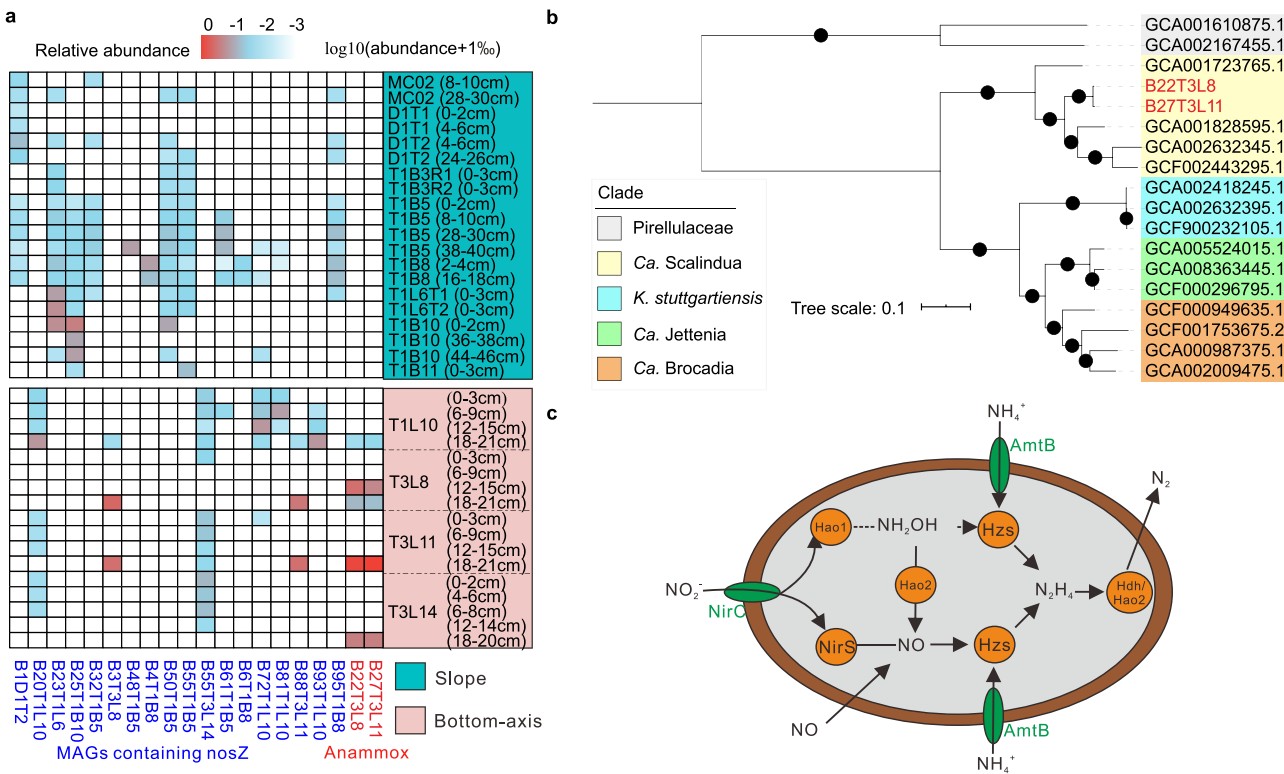

**Fig. 4 Identification of potential microbial nitrogen releasers in CD. a** Heatmap of the relative abundance of MAGs involved in anaerobic ammonia oxidation (anammox) and denitrification in sediments from the slope and bottom-axis sites. The relative abundance of each MAG was transformed using the logarithmic scale of 10 (log10(abundance + 1‰)) and was estimated using CoverM. **b** Phylogenomic analyses of 40 concatenated conserved proteins identified in anammox bacteria. Bootstrap values (1000 replicates) ≥80% are indicated by dots on branches. **c** Hypothetical nitrogen release in CD sediments via anammox. The anammox genes presented are from the CD bottom-axis sediment MAGs. Source data are provided as a Source data file.

rTCA (reductive Krebs cycle) were also detected at all examined hadal sites; however, a higher abundance of rTCA genes was observed in metagenomes from slope compared to bottom-axis sites. Bottom-axis metagenomes presented a higher abundance of genes involved in the Wood–Ljungdahl (WL) pathway compared to slope metagenomes (Supplementary Fig. 14 and Supplementary Discussion). While $CO_2$ fixation might be active at all sites, it is likely affected by available nitrogen and carbon pools, and microbial community composition at slope and bottom-axis sites. If complete oxidation of *n*-alkanes and aromatic compounds occurs in CD sediments, the produced $CO_2$ can possibly fuel the $CO_2$ fixers in the trench. However, this requires further investigation.

**Microbially mediated $N_2$ loss from CD bottom-axis sediments.** Our understanding of microbially mediated nitrogen cycling in CD sediments is still incomplete. We report two anammox MAGs affiliated to "*Candidatus* Scalindua" (MAG B22T3L8 and MAG B27T3L11) which dominated our four bottom-axis sediment cores below 12 cmbsf and were not present in the slope site samples (Fig. 4a). This suggests that the production of hadal nitrogen gas ($N_2$) via anammox can be different across the sites of CD trench. The anammox bacterial MAGs were mostly abundant in the anoxic 18–21 cmbsf sediment layer of T3L11 core from a bottom-axis site. These findings are consistent with the detection of "*Ca.* Scalindua" in the Ogasawara (9760 m)[11], Atacama (8,085 m) and Kermadec (10,010 m)[51] trenches, and the high abundance of anammoxers in deep sediments from the South China Sea (2610 m), identified using *hzs* and 16S rRNA gene cloning, respectively[11,52].

Phylogenetic analysis of 11 genes encoding hydroxylamine oxidoreductase (Hao)-like proteins identified in the two "*Ca.* Scalindua" MAGs shows that the identified (Hao)-like proteins clustered with Hao oxidoreductases and hydroxylamine hydrogenases (Hdh) taxonomically affiliated with Planctomycetota taxa known to be involved in anammox (Supplementary Fig. 15a). Four of our Hao-like proteins in the Hao2 clade clustered with hydroxylamine and hydrazine oxidation proteins that oxidize hydroxylamine ($NH_2OH$) to NO and hydrazine ($N_2H_4$) to $N_2$ in *Kuenenia stuttgartiensis*[53,54]. Two of our Hao-like proteins in the Hdh clade clustered with hydrazine dehydrogenases responsible for oxidization of hydrazine to $N_2$ in *K. stuttgartiensis*[53,54]. Three of our Hao-like proteins in the Hao1 clade clustered with candidate reductases capable of catalyzing $NO_2^-$ to $NH_2OH$ in "*Candidatus* Brocadia"[55,56]. Finally, the MAG B22T3L8 harbored genes that encode all subunits of hydrazine synthase (Hzs) that produces hydrazine using ammonium and nitric oxide as substrates for anammox (Supplementary Fig. 15b).

Other nitrogen-related genes identified in eight MAGs in bottom-axis samples included the nitrous oxide reductase gene (*nosZ*) responsible for the final step of denitrification and $N_2$ production (see Supplementary Discussion). Anoxic incubations of hadal sediments with [15]N-labeled compounds showed that the %$N_2$ production from anammox is different between trenches (Atacama Trench 67 ± 13%; Kermadec Trench >90%), but greater compared to what attributed to denitrification[51]. However, considering the different distributions of nitrogen producers (anammoxers/denitrifiers) and the available nutrient and oxygen pools across the CD sites and sediment depths, we speculate that nitrogen loss via anammox/denitrification is heterogeneously

distributed across the CD and might be intensified in bottom-axis sediments.

Finally, we did not identify any nitrogen fixation genes in CD hadal waters[30] or sediment metagenome (this study); however, this should be interpreted with caution considering that our MAGs encoded a high number of genes with no known homologs in public databases.

**Crosstalk of arsenic, selenium, and sulfur cycling for energy gain and detoxification in CD sediments**. Our measurements showed accumulation of arsenic and selenium in the CD sediments (Supplementary Fig. 1c, d). To reveal potential arsenic cycling in CD, we first searched our MAGs for the *ars* operon responsible for arsenate detoxification in bacteria[57,58] and archaea[59]. We found that 310/586 of the CD MAGs encoded at least one arsenate reductase (ArsC1/C2), while 162 MAGs encoded arsenite transporters (ArsB or Acr3) that remove the biotransformed arsenite out of the cytoplasm (Supplementary Fig. 16a and Supplementary Data 6). To our knowledge, genes related to arsenic detoxification/biotransformation in the hadal zone (*acr3, arsB*) have been reported only in hadal seawater metagenomes from the Yap hadal trench[24].

High As mobilization results from microbial activity in deep-sea sediments[60], and this might also occur in CD sediments. Indeed, we also identified that 385/586 CD MAGs harbored the *arsM* gene that produces methylated organoarsenicals. *arsM*, *aioA/arxA* (arsenite oxidation), and *arrA* (arsenate dissimilatory reduction) are involved in arsenic metabolism, and are less common in the environment compared to arsenic detoxification genes (e.g., *acr3, arsB, arsC*)[61]. Our data contradict this finding since *arsM* showed higher gene expression in all examined metatranscriptomes when compared to most commonly identified *ars* operon genes (Fig. 3c). The lack of available "omics" data describing arsenic cycling in other deep-sea and hadal sediments leaves only a comparison to available sediment data from the Guaymas Basin (<5000 m depth). We find that *arsM* genes show remarkably higher abundance in CD metagenomes when compared to metagenomes recovered from Guaymas sediments[27] (Supplementary Fig. 16b). However, interpretation of this difference is difficult because CD and Guaymas Basin have distinct geochemical conditions (e.g., presence of reducing hydrothermal fluids and intense sulfur cycling in Guaymas Basin[62]) that could also abiotically transform As[63]. We suggest that CD microbes recruit the *ars* operon detoxification strategy to overcome the potential toxic effects of arsenic accumulation in the sediments, but they also perform intense arsenic cycling that can provide additional survival advantages. These advantages might include piezoregulation in high hydrostatic pressures (>1000 atm) existing at the bottom-axis of CD, and/or production of primordial antibiotics involved in cell-cell combat as described elsewhere[17,18,58]. ArsM methylates arsenite to methylarsenite, a powerful trivalent organoarsenical, used as antibiotic-like compound by soil microbial communities[58]. Other organoarsenicals (e.g., AsB) resemble amino acid-based osmolytes[17,18], and may confer cryoprotection and protein stability under high hydrostatic pressures[64,65]. However, this requires further investigation considering that many organoarsenicals (including AsB) can be highly abundant in marine zooplankton and invertebrates and may derive from organic debris in the sediments[66,67].

We also detected expression of the *aioA* gene in our metatranscriptomes, which indicates alternative arsenic mobility via oxidizing arsenite to arsenate. Global-scale analyses of arsenic-related genes describe *arsM* and *aioA* as highly endemic genes in soil microbiomes that have limited regional dispersal compared to the cosmopolitan *ars* operon[61]. We suggest that this may also apply to the *arsM* and *aioA* detected in CD microbiomes considering the active arsenic cycling in CD sediments and the high microbial endemicity due to the topographic isolation of the trench. AioA-dependent chemoautotrophic arsenite oxidation is known to be coupled with denitrification for energy gain in a few bacterial representatives[68–70]. Arsenite oxidation with involvement of the *aioA* gene for chemoautotrophic growth has not been previously reported from deep-sea sediments but has been reported in 4–11% of microbial communities inhabiting the mesopelagic oxygen-deficient zone of Eastern Tropical North Pacific[71]. We propose that this activity may occur in CD microbes, especially those inhabiting the anoxic bottom-axis sediments where active denitrification and arsenic accumulation co-occur. Cryptic arsenic cycling may be more important than previously thought in biogeochemical cycling in global anoxic environments.

As mentioned, selenium also accumulates in CD sediments (Supplementary Fig. 1d) and we suggest that it is utilized by hadal microbes in CD for the synthesis of amino acid analogs (e.g., selenocysteine) and for arsenic biotransformation. *selA, selB, selD*, and *selU* genes coding for the selenocysteine synthase[72] were present in 49, 47, 50, and 5%, respectively, of our MAGs. Transcription of the *sel* genes involved in the synthesis of selenocysteine was detected in the three metatranscriptomes from the bottom-axis sample (Fig. 3c) (see also Supplementary Discussion). The Global Ocean Sampling project revealed that most prokaryotic selenoproteins of known function are oxidoreductases, thioredoxin (Trx)-like fold proteins, and homologs of known thiol oxidoreductases including arsenate reductases[73]. This suggests that CD microbes might have potential crosstalk between arsenic and selenium biogeochemistry that could favor arsenate detoxification as described elsewhere[72]. Our results also indicate that CD microbes can synthesize selenocysteine possibly as a result of an endemic CD adaptation, considering that biosynthesis of Sec is not a common trait among bacteria[74]. Addressing this hypothesis requires a refined search of hadal proteomes to avoid misinterpretations related to possible mis-annotations of selenoproteins[75] (see Supplementary Discussion).

Genes related to sulfur assimilation (*sat, cysH* and *sir*) were more abundant in our MAGs (Fig. 3a–c) compared with dissimilatory reduction genes (*sat, aprAB*, and *dsrAB*). The *sir/cysJ* genes involved in sulfite reduction were identified in 40% of our MAGs, compared to the 3.4% (666 MAGs in total) reported from other deep-sea sediments with active sulfur cycling (e.g., Guaymas sediments[27]). The *cysK* gene involved in cysteine synthesis was present in 50.7% of the MAGs (Supplementary Fig. 16c and Supplementary Data 7). Besides respiration and energy gain, evidence for sulfur cycling and cysteine biosynthesis in CD sediments could indicate the potential involvement of sulfur products in arsenic mobility and biodetoxification as described elsewhere[74,76,77] (Fig. 5 and Supplementary Fig. 16; see also Supplementary Discussion).

Overall, we generated metagenomes and metatranscriptomes from sediments collected at the deepest known hadal realm, the CD, to examine the biodiversity, metabolic capacity, and ecological roles of the microbiota across the trench (slope vs. bottom-axis sites). Our analyses indicate that sediment microbiota in CD can degrade a broad range of deposited organic compounds, and recycle macromolecules to sustain metabolic activity. CD microbes encode genes for denitrification and anammox that could contribute to hadal $N_2$ loss; however, the contribution of each process varies across the trench. Evidence of detoxification/utilization of heavy (As) and trace (Se) metals suggests that hadal sediment microbiota can retain types of ancient metabolisms for energy gain in CD. The high percentage of novel species identified in our study suggests that CD

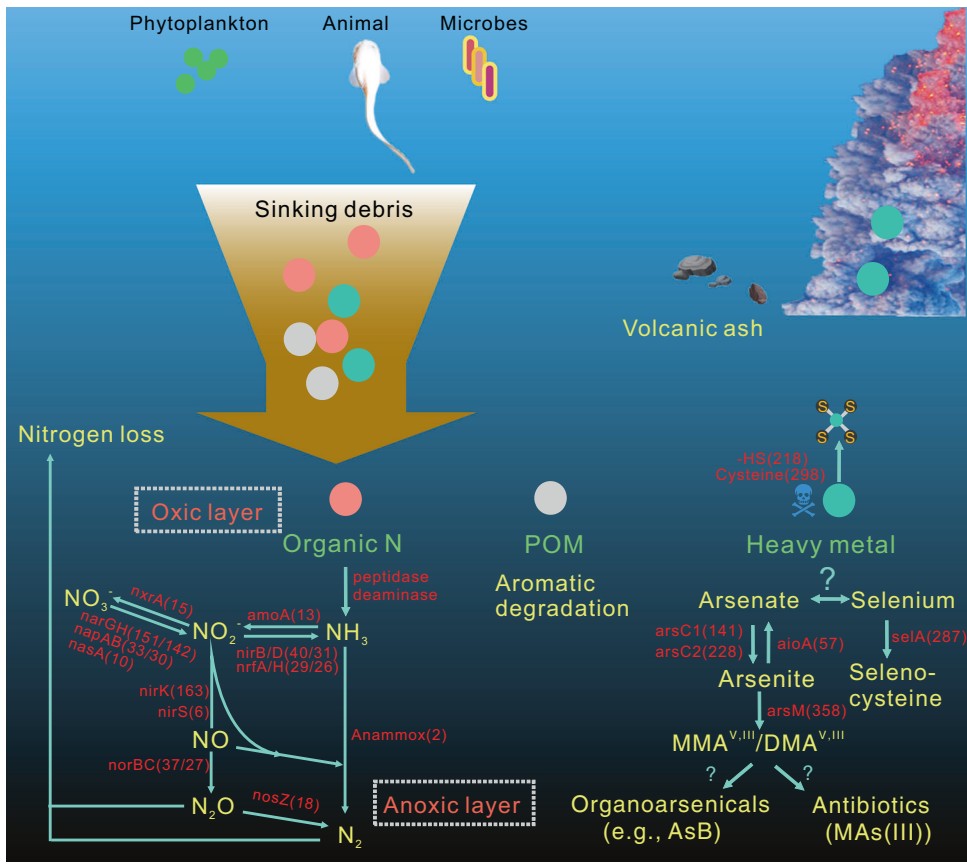

**Fig. 5 Schematic representation of potential nitrogen and arsenic cycling in bottom-axis CD sediments.** The number of genomes that contain the genes involved in nitrogen and arsenic cycling steps is in parentheses. Source data can be found in Supplementary Data 7.

sediments are large reservoirs of poorly known microbes that contribute to deep ocean biogeochemistry and may be mined for biotechnological research and applications.

## Methods

**Sampling and geochemical analyses.** Thirteen CD sediment cores were collected from both slope and bottom-axis sites at depths ranging from 5400 to 10,911 m using different methods (Fig. 1a; Supplementary Fig. 2) during three hadal cruises *R/V Dayang 37-II* (DY37II, June–July of 2016), Tansuo01 (TS01, June–August of 2016), and Tansuo03 (TS03, January–March of 2017). Eight cores were collected by push core and five were collected by box core (Supplementary Data 2). All intact core samples were immediately subsampled into 2- or 3-cm layers (Supplementary Data 2) onboard at room temperature and were then frozen at −80 °C until use. Photos of samples and push/box core are shown in Supplementary Fig. 2. The temperature of the sampling sites was recorded using a conductivity, temperature, and depth sensor (Seabird, Bellevue, WA, USA).

Sediment samples were centrifuged at $2000 \times g$ for 15 min to collect the porewater (~10 m), which was filtered using a 0.22-μm syringe filter (Millipore, Bedford, MA, USA) for nutrient analysis. Porewater nutrient concentrations ($NO_3^-$, $NO_2^-$, and $NH_4^+$) were analyzed onshore using an automated continuous-flow SEAL AA500 analyzer spectrophotometer (SEAL Analytical, Germany). 5–10 g of CD sediments were freeze-dried and powdered with mortar and pestle. 0.1–0.2 g of the homogenized sample was mixed with 10 ml of Aqua Regia solution used as a digestive agent following the Chinese Standard (GB17378.5-2007) for analysis of arsenic (As) in marine sediments. The total As in the digested samples was quantified using atomic fluorescence spectrometry (ASF930, Tian Instruments, Beijing, China). 0.2–0.3 g of the homogenized sediments were mixed with a solution of 5 ml 0.1 M ascorbic acid and 0.3 M phosphoric acid in a polytetrafluoroethylene (PTFE) centrifuge tube and were sonicated for 10 min. The arsenic species were extracted from the liquid phase and separated by centrifugation ($1301 \times g$ for 20 min). These treatments were performed twice. The supernatants were combined and filtered through a PTFE membrane filter (0.22 μm; Millipore, Bedford, MA, USA cat# FGLP04700). The As compounds (As(III), As(V), MMA, and DMA) were separated using high-performance liquid chromatography (HPLC; BSA-100, Baode, China) and then quantified by atomic fluorescence spectrometry (BAF-2000, Baode, China). Total selenium and mercury were analyzed following the Chinese Standard (HJ680-2013). 0.5 g (±0.0001 g) of

the homogenized sample was mixed with 6 ml of 12 M hydrochloric acid (Guangzhou Chemical Reagent Factory, Guangzhou, China cat#CB11-TD) and 2 ml of 16 M nitric acid (Guangzhou Chemical Reagent Factory, Guangzhou, China cat#CD18) solutions. The mixed solution was heated for 5 min at 150 °C for digestion and then transferred to a 50-ml volumetric flask. The total selenium and mercury were quantified using atomic fluorescence spectrometry (BAF-2000, Baode, China; HPLC, BSA-100, Baode, China). The total organic carbon and total nitrogen content of the sediments were measured by flash combustion using a Flash EA 2000 series elemental analyzer (Thermo Fisher Scientific, Waltham, MA, USA). The sediment samples were acidified for 24 h with 10 ml of 1 M HCl prior to the analysis, to remove the inorganic carbon content, and were then freeze-dried and powdered.

**Nucleic acids extraction, metatranscriptome, and metagenome library preparations.** We selected 37 sediment samples from the different slope and bottom-axis sites from different water depths (Supplementary Data 2). DNA was extracted from 10–40 g of sediment using the PowerMax soil DNA isolation kit (MoBio, Carlsbad, CA, USA cat#12988-10), according to the manufacturer's instructions. DNA concentrations were measured using a Qubit™ 2.0 Fluorometer (Invitrogen, Carlsbad, CA, USA cat#Q32851). Samples with DNA concentration of ≤2 ng/μl were concentrated using AMPure XP beads (Beckman Coulter, CA, USA cat#A63881) prior to library preparation. A total amount of 100 ng genomic DNA was used to prepare libraries with an insertion size of 550 bp or 350 bp using the TruSeq Nano DNA LT Library Prep Kit (Illumina, California, USA cat#FC-121-4002). Two control samples (blanks) were processed following the same DNA extraction protocol to detect potential DNA contamination due to handling and/or kit reagents. The DNA from each blank was concentrated using AMPure XP beads (Beckman Coulter, CA, USA cat#A63881) and yielded <2 ng in a total volume of 10 μl. The amount of DNA in the control samples did not meet the minimum requirement of the TruSeq Nano DNA LT Library Prep Kit (Illumina, California, USA cat#FC-121-4002), which was used for the preparation of the sediment libraries. For this reason, we used the TruePrep DNA Library Prep Kit V2 for Illumina kit (Vazyme, Nanjing, China cat#TD503) to prepare the control libraries with an insertion size of 350 bp.

RNA from three sediment layers (6–9, 12–15, and 18–21 cmbsf) of the bottom-axis sediment sample collected from the T3L11 site (10,908 m depth) was extracted from ~10 g sediment using a PowerSoil Total RNA Isolation Kit (MoBio, Carlsbad, CA, USA cat#12866-25) following the manufacturer's instructions. We performed

RNA extractions in triplicate from each sample. The pooled RNA was concentrated using one RNA binding column and then quantified using a Qubit™ 2.0 Fluorometer (Invitrogen, Carlsbad, CA, USA cat#Q32852). To ensure DNA removal, the RNA extracts were treated with TURBO DNase (Invitrogen, Waltham, MA, USA cat#AM2238) as directed by the manufacturer. The remaining RNA was used as a template for a PCR with universal primers 341F (5′-CCTAYGGGRBGCASCAG-3′) and 802R (5′-TACNVGGGTATCTAATCC-3′) to ensure removal of potential DNA carryover. Each 50 μl PCR reaction contained 1.25 U PrimeSTAR HS DNA Polymerase (Takara, Japan cat#R010B), 5× PrimeSTAR Buffer (Takara, Japan, cat#R010B), 200 mM dNTPs (Takara, Japan, dNTP Mixture, cat#R010B) and 0.3 μM of each primer (final concentrations). These reactions were performed at 94 °C for 10 s, followed by 35 cycles of 98 °C (10 s), 55 °C (10 s), and 72 °C (30 s). We used the Ovation® RNA-Seq System V2 Kit (NuGEN, San Carlos, CA, USA cat#7102) to convert 1 ng RNA into cDNA (ds-cDNA), and prepared the metatranscriptome libraries as described in the metagenome library preparation.

**"Omic" data sequencing and assembly**. The DNA/cDNA libraries were sequenced using Illumina Miseq 2 × 300, Novaseq 6000 2 × 150, or Hiseq 2000 2 × 150 flow cell (Supplementary Data 3). Raw reads were trimmed to remove adapters and then filtered using fastp (v.0.20.0)[78] with parameters (-w 16 -q 20 -u 20 -g -c -W 5 -3 -l 50). Low quality reads (assigned by a quality score <20 for >20% of the read length), those shorter than 50 bp and unpaired were removed. Reads that mapped to the in-house contaminant database (including sequences from higher eukaryotes and common laboratory contaminant bacteria genomes[79] downloaded from NCBI) using Bowtie2 (v.2.4.1)[80], and the rRNA sequence database by SortMeRNA (v.2.1)[81] were discarded. Detailed information about samples and corresponding sequencing data are shown in Supplementary Data 3. The qualified reads for each site (MC02, D1T1, D1T2, T1B3, T1B5, T1L6, T1B8, T1B10, T1B11, T1L10, T3L8, T3L11, and T3L14) were merged for assembly using SPAdes (v.3.13)[82] with a kmer set of 21, 33, 55, 77, 99 and 127 under the '--careful' mode[30] to achieve the best assembly results for low-abundance microbial groups. Before binning, scaffolds <2000 bp were removed.

**Estimation of microbial abundance and microbial genome novelty**. Bowtie2 (v.2.4.1) was used (with setting -N 1 --un-conc) to align reads of 37 sediment samples to our co-assembly (>300 bp) of two control metagenomes. Reads that mapped to the control co-assembly were removed from further analysis and >99% of reads remained in 34 sample datasets (Supplementary Data 3). Ribosomal RNA tags (5S, 16S, and 23S) were extracted from qualified metagenomic reads using rna_hmm3.py[83], which employed HMMER (v.3.1b2)[84] to predict ribosomal RNA gene fragments from both forward and reverse metagenomic reads. An in-house python script was used to extract 16S and 18S miTags (≥100 bp), respectively (https://github.com/ucassee/Challenger-Deep-Microbes)[85]. The 16S and 18S miTags were imported into Qiime2 (v.2019.7.0)[86] with the setting of --type 'SampleData [Sequences]' and dereplicated redundancy to generate representative sequences. Classify-consensus-vsearch command in Qiime2 was used to classify the representative miTags in reference to SILVA 132 SSU database.

The 16S miTag sequences were searched by BLASTn (v.2.9.0) against SILVA 138 SSU database, using 97% as a threshold to identify novel 16S miTags at the species level. Novel 16S miTags rates were calculated by the number of novel 16S miTags identified in each sample divided by the total number of 16S miTags that each sample contained. 16S miTag sequences for deep-sea sediments and CD water samples were extracted from public metagenomes (Data sources and identity results were deposited in https://doi.org/10.6084/m9.figshare.12979709). MNS[32] was calculated by submitting classification and abundance results from Parallel-META 3[87] to the microbiome search engine webtool (http://mse.single-cell.cn/index.php/mse). Public metagenome data were downloaded from NCBI using prefetch (v.2.1.5, https://github.com/ncbi/sra-tools).

**Genome binning and decontamination**. Genome binning was performed on each assembly (13 individual sampling sites and 2 controls) by running three different tools MaxBin (v.2.2.6)[88], MetaBAT (v.2.12.1)[89] and CONCOCT (v.1.0.0)[90] using their default settings. MaxBin[88], MetaBAT[89], and CONCOCT[90] bin the genomes based on a combination of sequence composition (tetranucleotide frequency) and coverage level. Raw genome bins resulting from these three binning approaches were combined, and the best binning result was selected for each genome set using the bin_refinement module in metaWRAP (v.1.2.2)[91]. During the bin refinement process, we used CheckM_lineage (v.1.0.12)[92] to evaluate the percentage of completeness and contamination for each bin. Good quality MAGs (≥50% completeness and ≤10% contamination) were retained for further analysis. This yielded a total of 645 bacterial and 41 archaeal MAGs. Next, we used dRep (v.1.4.3)[93] to choose the best representative genome for each genome set to dereplicate redundant MAGs with setting "-comp 50 -con 10." This resulted in 590 prokaryotic MAGs. We identified in our control MAGs contaminant species using the Genome Taxonomy Database (GTDB), and thus we removed four bacterial MAGs. This led in a total of 586 MAGs that were used for downstream analyses.

**Phylogenetic analyses**. We used GTDB-Tk (v.1.0.2)[94] to taxonomically classify our MAGs with the GTDB release 89[36]. An in-house python script was used to pick the most closely related genomes in GTDB release 89. These genomes were combined with genomes from NCBI, which resulted in a manually curated reference genome set (n = 255, Supplementary Data 8) for further analysis. SpecI (v.1.0)[95] was used to extract 40 universal, single-copy phylogenetic marker genes (Supplementary Data 9). In cases where the selected marker genes had more than one gene copy in the genome, we retained the gene copy that encoded the protein with the highest HMM bit score. Mafft (v.7.407)[96] with setting --maxiterate 1000 --localpair was used to independently align protein sequences of the marker genes. We applied trimAl (v.1.4)[97] to further trim the poorly aligned regions in each aligned conserved protein. Alignments of 40 conserved proteins were concatenated and missing proteins in an MAG were regarded as alignment gaps. Finally, the aligned columns having gaps and ambiguous amino acids in >50% of genomes were removed by an in-house python script, which resulted in a final protein alignment consisting of 8107 aligned columns. The maximum-likelihood (ML) phylogenomic tree was constructed using the concatenated aligned protein sequences with IQ-TREE[98] tool (setting: -m MFP -asr -bb 1000 -nt AUTO), and the best-fit substitution model (LG + R10 model) for 1000 replicates.

We used phylogenetic analysis to characterize other key genes of interest (i.e., *hao* and *nxrA*). These genes were identified in MAGs using KofamScan, GhostKOALA, or DIAMOND (for details see below). Published reference proteins encoded by these genes were downloaded from NCBI-nr database and UniProt Knowledgebase (UniProtKB). Protein sequences from MAGs and reference databases were combined and aligned using MUSCLE, and trimmed using trimAl with the settings described above. The ML trees for the proteins were constructed by raxmlHPC-PTHREADS-AVX (v.8.2.12)[99] with settings of -x 12345 -p 12345 -f a -# 1000 -T 28 -m PROTGAMMAILG. All phylogenetic trees were visualized using the interactive Tree Of Life (iTOL v.4) tool[100].

**Genome abundance and prevalence**. The estimation of the relative abundance in MAGs depends on the recruitment rates of the qualified reads[101]. Qualified metagenomic reads were aligned to all MAG contigs using bwa mem (v.0.7.17)[102] with the default parameters and sorted by samtools (v.1.9)[103]. The identified reads with an alignment higher than 99% identity and a minimum alignment length of 50 nt, after the quality control and dereplication of the MAGs, were considered as a good hit. Relative abundance of MAGs was estimated based on genome coverage and number of reads by relative_abundance method in CoverM (v.0.4.0, with setting -m relative_abundance -min-read-aligned-length 50 --min-read-percent-identity 0.99 --min-covered-fraction 0.1 --proper-pairs-only in genome mode) (https://github.com/wwood/CoverM) using the following equation:

$$\text{Genome relative abundance} = \frac{C1 \times R1}{C2 \times R2} \tag{1}$$

where C1: genome mean coverage, C2: sum of mean coverage of all MAGs, R1: the number of reads mapped to all MAGs, and R2: number of all metagenomic reads. We estimated the abundance of MAGs that presented a min-covered-fraction of >10%. The abundance of the MAGs in each metagenome was imported to "*vegan*" package (v.2.5-4) in R to estimate Bray–Curtis dissimilarity index for PCoA ordination analysis.

To determine the most transcriptionally active species in the three metatranscriptomes, we clustered the sediment MAGs using an Average Nucleotide Identity threshold of 95%, and we used the clustering outcome to recruit the metatranscriptome reads. The number of reads mapped to each MAG cluster was divided by the total number of the mapped reads to estimate the transcriptionally active species. The results were visualized using the heatmap package of R platform (v.3.6)[104].

**Reads and genes annotations**. Clean metagenome reads were annotated with DIAMOND[105] BLASTx (v.0.9.27.128, -c 1 -daa) against NCBI-nr database (2019-10), and then using daa2rma (-ms 50 -me 0.01 -top 50) and rma2info (-r2c SEED -n true --paths true --list true --listMore true -v) tools in MEGAN (v.6.18.5) with megan-map-Oct2019.db to parsed the NCBI-nr annotations into SEED categories.

Proteins for individual MAGs were predicted using prodigal[106] (v.2.6.3) with option "-p meta." Using KofamScan (v.1.1.0)[107] and GhostKOALA (v.2.2)[108], KEGG Orthologs (KOs) in release 92.0 were assigned to protein sequences. For a detailed module and pathway analysis, the KO numbers were downloaded, concatenated, and merged with a KO-to-pathway metadata file using a python script (https://github.com/ucassee/Challenger-Deep-Microbes)[85]. We used dbCAN2[40] to annotate carbohydrate degradation enzymes and DIAMOND[105] BLASTp (e-value < 1e−10, sequence identity >50% and the shortest alignment rate >80%) to search against Merops database (release 12.1). To identify novel proteins, eggNOG-mapper (v.2.0.0) was used against eggNOG[109] with default setting -m diamond --seed_ortholog_evalue 1e−5 and DIAMOND[105] BLASTp (e-value <1e−5, sequence identity >40% and the shortest alignment rate >40%) was used to search against NCBI-nr database.

**Function abundance profiling**. The SEED functional profiles of metagenomes were estimated by the number of reads classified to each SEED class divided by the number of all annotated reads (see section "Reads and genes annotations"). One

million reads were randomly picked by seqtk (v.1.3, https://github.com/lh3/seqtk) from forward reads of each metagenome to reduce the running time of DIAMOND[105] BLASTx (v.0.9.27.128) for reads annotation against the SEEE database. To calculate the relative abundance of functional genes, we also mapped clean reads from the 37 metagenomes and the three metatranscriptomes to all gene-coding nucleotide sequences in our MAGs, with the same method described in genome relative abundance estimation. The aligned reads were further filtered and estimated using CoverM (v.0.4.0, >50 bp alignment length, >95% identity for metagenomes and >97% identity for metatranscriptomes in contig mode). The abundance of each gene in metagenomes and metatranscriptomes was calculated as the metric-TPM (transcripts per million) normalized based on the gene length and sequencing depth using the following equation and an in-house python script:

$$\text{TPM}_i = \frac{R_i}{L_i} \cdot \left( \frac{1}{\sum_j \frac{R_j}{L_j}} \right) \cdot 10^6 \tag{2}$$

where $R_i$ represents read counts for the specific gene and $L_i$ is the gene length (kb). We subsequently summed the TPM of genes that were classified to the same function to calculate the functional abundance of each meta-omics library.

**Reporting summary**. Further information on research design is available in the Nature Research Reporting Summary linked to this article.

## Data availability

The raw metagenome and metatranscriptome sequencing data and MAG sequences generated in this study have been deposited in the NCBI database under the accession code PRJNA635214. The processed gene annotation and 16S miTag sequences are available at Figshare (https://doi.org/10.6084/m9.figshare.12979709)[110]. Accession numbers for the previous published metagenomes and MAGs used for comparisons can be found in the Source data file of Fig. 1b and in the NCBI database under accession code PRJNA362212. Public available datasets used in this study include the checkM v.1.0.12 database (https://data.ace.uq.edu.au/public/CheckM_databases/), the GTDB database release 89 (https://data.gtdb.ecogenomic.org/releases/release89/), the SILVA 132 and 138 SSU database (https://www.arb-silva.de/download/archive/), HMM profiles of KEGG release 92.0 (https://www.genome.jp/ftp/db/kofam/archives/), MEROPS database release 12.1 (https://www.ebi.ac.uk/merops/download_list.shtml), dbCAN2 database (https://bcb.unl.edu/dbCAN2/download/Databases/), megan-map-Oct2019 database (https://software-ab.informatik.uni-tuebingen.de/download/megan6/old.html), and eggNOG 5.0 database (http://eggnog5.embl.de/download/eggnog_5.0/). Correspondence and material requests should be directed to Y.W. (wangyong@sz.tsinghua.edu.cn). Source data are provided with this paper.

## Code availability

The custom scripts used in this study are publicly available at GitHub (https://github.com/ucassee/Challenger-Deep-Microbes) and Zenodo (https://doi.org/10.5281/zenodo.6061243)[85].

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

## Acknowledgements

We give special thanks to the members of the R/V DY37, TS01, and TS03 for their invaluable efforts in the sampling cruises. We thank J. Li, S.X. Wang, Y.Z. Xin, J. Chen, and D.S. Cai for their skillful handling of the lander and sediment sampler. We also thank X.Y Song's laboratory for their help in porewater nutrient analyses. This study was supported by the Strategic Priority Research Program B of the Chinese Academy of Sciences (No. XDB06010201 to Y.W.).

## Author contributions

Y.-L.Z. and Y.W. conceived the study. G.-J.C. and Y.-L.Z. performed the experiments. Y.-L.Z. and P.M. analyzed data and summarized the results. Y.-L.Z. and P.M. wrote the manuscript. V.P.E. and Y.W. critically revised the manuscript.

## Competing interests

The authors declare no competing interests.
