## [Peer Review File · Nature Communications]

REVIEWER COMMENTS

Reviewer #1 (Remarks to the Author):

The paper presents an analysis of a large number of metagenome-assembled genomes (MAGs) from the Challenger Deep and its flanks in the Mariana Trench. The analysis focuses on the metabolic functions represented in these genes and in this way aims at improving our understanding of benthic microbial ecosystems in the deepest part of the oceans.

Hadal trenches were recently shown to be hotspots of microbial activity in the deep sea. Together with recent developments in instrumentation, this has made the biogeochemistry and microbial ecology of trench sediments a hot topic of wide interest and spurred numerous investigations. To date, few microbial genomes are available from hadal trenches, and the present study thus has the potential to provide new perspectives in its field.

The manuscript presents a very large dataset of high quality, and it represents a very large effort. Still, and surprisingly, the conclusions drawn are vague and do not seem like a major step forward in our understanding of hadal microbial ecology. The manuscript lacks focus and clear research questions, it does not clearly identify novel insights nor discuss these in the perspective of previous findings, and it does not give proper credit to earlier work. To exemplify this, I quote the major conclusions from the abstract:

1) "...hadal microbes are potentially active heterotrophic degraders and ammonia oxidizers...". I fail to see the novelty in this, particularly considering the use of "potentially" here. Previous studies have shown that hadal microbial communities are highly diverse and contain lineages that are known as heterotrophs, and that their activity is driven by the flux of organic matter, i.e., they must be active heterotrophs. Previous studies have also shown high abundances of ammonium oxidizing Archaea in hadal sediments including the Challenger Deep (Nunoura et al. 2018, doi:10.1264/jsme2.ME17194, not cited).

2) "...and likely capable to reduce arsenate and selenate possibly as a detoxification and/or energy gain strategy." Again, the conclusion is weakened by "likely" and "possibly" and the relevance is not obvious without data on arsenate and selenate.

3) "We report for the first time anammox and denitrifying bacteria present in CD sediments indicating the importance of hadal microbes in deep-sea nitrogen cycling,...". Anammox, denitrification and nitrification were already reported from a hadal site by Nunoura et al. in 2013 (doi:10.1111/1462-2920.12152, not cited), and nitrogen cycling was also demonstrated by this group in the Challenger Deep (above ref.). Thus, the conclusion is overstated and the novelty is not clear.

4) "Our results ... provide the first data on functional characterization of microbiota in hadal sediments." Clearly this is incorrect considering the work by Nunoura et al. mentioned above, but also, e.g., in view of recent publications by Peoples and coworkers as well as a host of earlier cultivation-based studies.

The dataset described here could be a gold mine, but the value does not come across to the reader. My recommendation is either to expand the paper to a longer format which can be presented in a more specialized journal, or to focus on one or a few aspects which can be treated coherently in proper depth and detail with robust conclusions and clear delineation of the novelty and perspectives for our understanding of the hadal realm. In the present form, the paper covers too many aspects superficially without tying them together, and with the lack of clear research questions, it does not appeal to the broad audience of a generalist journal such as Nature Communications.

Reviewer #2 (Remarks to the Author):

Zhou et al. present a study of metagenomes and some metatranscriptomes recovered from sediments within the axis of the Challenger Deep as well as from north or south trench slope.

There are many things to recommend this study. In addition to overcoming all the technical challenges associated with obtaining the sediments the authors have also obtained extensive geochemical data and also have some metatranscriptomes, so that descriptions of activity are also possible. The statistical analyses are robust, the figures are nicely prepared, the bioinformatics methods are well described and in general the materials and methods are excellent, with some exceptions noted below. However, the take home messages from this work do not seem to be particularly surprising. Finding new features for the first time in the microbial populations of the Challenger Deep (CD) is certainly important to specialists in this habitat, but are their discoveries of more general fundamental significance?

Line 61- This suggests localized evolution as opposed to evolution elsewhere and transport to the CD. This would seem to be refuted by the high 16S similarity among microbes in different benthic hadal and non-hadal settings. Perhaps a hypothesis to evaluate with the MAGs obtained through this study?

Line 114- Any eukarya present? An overview of their diversity could also be briefly described.

Line 124 - The percentages of novel 16S miTags is higher from the CD than from vent and seep sediments and also in bottom-axis versus slope sediments. This is very interesting. What about in comparison to sediments from other water column depths such as abyssal or bathyal?

Line 183 and suppl. Fig 5. The metatranscriptomes are not extensively evaluated in this paper. They don't seem to contribute much to the story.

Line 203 - Distinct MAGs are described to be present in the slope and bottom-axis, and many others are shared between these two sample types. This is also interesting. But, this study does not take the analyses to the next level and address the kinds of distinct microbes/functions that distinguish these 3 groups?

Line 220 - the functional characterizations become tedious. This is great stuff for the trench sediment microbiome aficionados, but a slog for others. What is needed is the thirty thousand feet view of the

functions present here (or not) in comparison to those present in other marine sediments. Is it really necessary to include descriptions of glycolysis, C3 and C4 metabolite interconversions, and amino acid metabolism in the main body of this paper? If so it should be explained. It does strike me that the pupylation story and its connection to protein turnover in eukarya is valuable.

Line 339. In a similar vein the nitrogen story would benefit from a bigger picture perspective. What is unexpected? Finding anammox in the CD is new, but if it was a surprise then why? Are there aspects of the anammox communities in the CD or their associated geochemical profiles and gene distributions that are distinct from those present in other marine sediments?

Line 384 - Finding genes associated with arsenic detoxification seems novel - is it? Were the arsenic levels (or those of other heavy metals) high?

Line 399 - what distinguishes figure 6 details from other marine sediments?

Line 402. An integrated results and discussion would make it easier for readers to connect the results with their significance.

Line 517 - More information on the sample recoveries is needed. The supplementary data 2 table should describe how samples not collected by lander or submersible were collected - presumably hydrographic cable. Some description of lander pushcore operation is also needed in the supplementary information document or in the main materials and methods. Perhaps the lander is described in another publication? Information on length of time required to bring sediment cores on the ship, the temperature of the recovered cores, and their processing conditions (e.g., room temp or not) would be helpful.

Line 529 - It is indicated that analysis included arsenic. What else was measured? This additional data should be included.

Line 530 - what information is present in the parentheses?

Line 539. The metagenome data is described in detail in the paper, but the same is not true for the metatranscriptome data. Its quality and quantity should be described. How can one be confident that the cDNAs generated reflect expression in situ? Perhaps from the types of microbes and genes showing highest expression levels?

=====

-What are the noteworthy results?

13 sediment samples were recovered, some from the deepest marine sediment locations known, and 37 metagenomes and 586 MAGs were described. Anaerobic ammonia oxidation has been found to occur at the deepest ocean location.

- Will the work be of significance to the field and related fields? How does it compare to the established literature? If the work is not original, please provide relevant references.

While hadal research has blossomed in recent years, this study provides the most detailed analyses to date of the functional properties of microbes in Challenger Deep sediments. The analyses performed here are in many cases more rigorous than those previously published.

- Does the work support the conclusions and claims, or is additional evidence needed?

Yes.

- Are there any flaws in the data analysis, interpretation and conclusions? - Do these prohibit publication or require revision?

No.

- Is the methodology sound? Does the work meet the expected standards in your field?

Yes.

- Is there enough detail provided in the methods for the work to be reproduced?

Yes, except in a few places which have been noted.

Reviewer #3 (Remarks to the Author):

The researchers used sediment collected from 13 slope and axis sites of the Challenger Deep to generate 37 metagenomes from which 586 MAGs were assembled. Metatranscriptome data were also obtained for three samples. The data allowed the authors to survey the potential metabolic capacities and the likely ecological roles of sediment microbes in the Deep. The dataset is of good quality, interesting, and publishable. The manuscript provides new insights into the microbial inhabitants and their potential contributions to biogeochemical cycles (e.g. N, As) and life processes in the hadal biosphere. However, I feel that the manuscript did not make me so excited about the work for the following reasons.

(1) First, we already know quite a bit about the hadal biosphere, for example, microbial oxidation of hydrocarbons. What are the new discoveries in this study that make it stand out?

(2) Second, I was curious about if the authors have found any evidence for eukaryotic microbes, e.g., fungi in the hadal sediment.

(3) Finally, the hypothesis presented in the manuscript was not exciting and offers no intrinsic aspirations to the reader to further explore the hadal biosphere.

(4) One other point is that the authors failed to discuss any ecological or physiological implications for the discovery of genes encoding for arsenate reduction/detoxification. This can be an important selling point for this work.

(5) I would also suggest that a comparison and contrast in microbial communities and their metabolism be discussed between the water column and sediment in the Mariana Trench as such data is available in

the literature.

(6) The statement line 418-423 is confusing. Genome size for MAGs unique to the slope or axis is smaller than those shared between slope and axis? How does this offer an advantage to microbial adaptation in the trench?

(7) There are a number of syntax errors in the manuscript, e.g., line 415-417.

Given these problems, I recommend major revisions before being considered for publication in Nat. Comm.

We thank the reviewers for their constructive comments and for recognizing the unique nature of this data set and the contributions it makes to our understanding of hadal ecosystems. We feel that we have been able to address the comments, and that these changes have greatly improved the manuscript. Thank you. Please see our responses below:

Reviewer #1 (Remarks to the Author):

1. The paper presents an analysis of a large number of metagenome-assembled genomes (MAGs) from the Challenger Deep and its flanks in the Mariana Trench. The analysis focuses on the metabolic functions represented in these genes and in this way aims at improving our understanding of benthic microbial ecosystems in the deepest part of the oceans.

Hadal trenches were recently shown to be hotspots of microbial activity in the deep sea. Together with recent developments in instrumentation, this has made the biogeochemistry and microbial ecology of trench sediments a hot topic of wide interest and spurred numerous investigations. To date, few microbial genomes are available from hadal trenches, and the present study thus has the potential to provide new perspectives in its field.

The manuscript presents a very large dataset of high quality, and it represents a very large effort. Still, and surprisingly, the conclusions drawn are vague and do not seem like a major step forward in our understanding of hadal microbial ecology. The manuscript lacks focus and clear research questions, it does not clearly identify novel insights nor discuss these in the perspective of previous findings, and it does not give proper credit to earlier work. To exemplify this, I quote the major conclusions from the abstract:

Response: We thank the reviewer for these insightful comments. We believe that the revised manuscript is now improved following this reviewer's critical suggestions. The pioneering works in Ogasawara trench and Mariana trench, that we mistakenly did not cite before, (we apologize for that), are now cited and discussed (see lines 57-62, 80-85 and 349-352). Regarding N cycling, we revised our manuscript (starting line 328 and Supplementary Discussion line 119) to further emphasize the likely importance of N loss from these hadal trench sediments (and potentially other hadal sediments) to global ocean N cycling. We added results (starting line 381) to our revised manuscript about cryptic arsenic cycling mediated by hadal microbes. Upon the suggestions of the other two reviewers, we searched further into our 37 metagenomes from 13 sites along the V-shaped trench to provide more information on the distribution of selected pathways and we now show a much higher level of novelty in our microbiomes, with capacity for heavy metal detoxification (one of the main findings of our research). Also, to support further our 'omics' data with regards to heavy metal biotransformations/detoxification, we added measurements of total arsenic, arsenate,

total mercury, and total selenium (now included in the revised manuscript lines 104-113 and Supplementary Fig. 1c-e and Supplementary Data 1) from the deepest site that we sampled (10,911 m).

2. *“...hadal microbes are potentially active heterotrophic degraders and ammonia oxidizers...”. I fail to see the novelty in this, particularly considering the use of “potentially” here. Previous studies have shown that hadal microbial communities are highly diverse and contain lineages that are known as heterotrophs, and that their activity is driven by the flux of organic matter, i.e., they must be active heterotrophs. Previous studies have also shown high abundances of ammonium oxidizing Archaea in hadal sediments including the Challenger Deep (Nunoura et al. 2018, doi:10.1264/jsme2.ME17194, not cited).*

Response: Thank you for this comment. As we mentioned on our previous response, we failed to cite the Nunoura et al., papers which are now included in our Introduction (lines 59, 61 and 85) as well as in our Results and Discussion (lines 349-355, and please see also our response to comment #4). We agree with the reviewer that Nunoura et al, 2018 showed ammonia oxidation; however, the authors discussed evidence only for aerobic ammonia oxidation in Challenger Deep (*amo* gene), while the unsuccessful amplification of key anammox genes (e.g., *hzsA*) led the authors to state “suggests that anammox is not a major player in the nitrogen cycle in the trench bottom sediments (Nunoura et al., 2018)”. Here we reveal two MAGs from anammox bacteria that dominated below 12 cmbsf (anoxic sediment layers) in the bottom axis. We believe that our findings of a complete anammox pathway in the anammox MAGs together with evidence for actively transcribed genes involved in denitrification at the anoxic CD sediments, can contribute to N₂ loss from the hadal sediments to the abyssal water column. See also our reply to this reviewer’s comment #4.

We apologize for the confusion in the sentence of “...hadal microbes..” in our abstract. AOA had been detected in our previous study as well, so it has no novelty. As now noted more clearly in our abstract, what this manuscript shows to readers is that hadal microbiota are engaging in metabolisms involved in heavy metal detoxification and N loss from the trench, the latter of which has never been reported on the basis of such a large number of samples and metagenomes.

3. *“...and likely capable to reduce arsenate and selenate possibly as a detoxification and/or energy gain strategy.” Again, the conclusion is weakened by “likely” and “possibly” and the relevance is not obvious without data on arsenate and selenate.*

Response: Thank you for this suggestion. This sentence in our abstract now reads: “The capacity of the microbes to reduce arsenate and selenate as a detoxification and/or energy gain strategy is supported by the significant presence, and the enriched transcription of genes related to arsenical biotransformation in our MAGs and metatranscriptomes, respectively”. To support our ‘omics’ findings we now add measurements of arsenate, total arsenic, mercury and selenium in our sediment samples,

now discussed in our revised manuscript (starting line 104 and please see Supplementary Fig. 1c-e). Ideally, we would have measured other heavy metals of interest in Marianna Trench which appear microbially detoxified (e.g., Cd, Pb, etc); however, these types of analyses require a significant amount of sediment, and thus we were restricted to selecting only those heavy metals for which we had the most evidence for utilization in our 'omics' data. With these new measurements, however, we provide evidence of presence of heavy metals in the CD sediments, and this makes us more confident in our conclusions about heavy metal biotransformations for energy and/or detoxification. We now provide a more detailed discussion about crosstalk between arsenic, selenium and sulfur now starting at line 380 (section "*Crosstalk of arsenic, selenium and sulfur cycling for energy gain and detoxification in CD sediments*").

4. "*We report for the first time anammox and denitrifying bacteria present in CD sediments indicating the importance of hadal microbes in deep-sea nitrogen cycling, ...*". *Anammox, denitrification and nitrification were already reported from a hadal site by Nunoura et al. in 2013 (doi:10.1111/1462-2920.12152, not cited), and nitrogen cycling was also demonstrated by this group in the Challenger Deep (above ref.). Thus, the conclusion is overstated and the novelty is not clear.*

Response: Thank you for this comment. The Nunoura et al. (2013; 2018) pioneer works are now cited. We agree with the reviewer and thus we rephrased as follows "We report MAGs from anammox bacteria dominant in CD bottom-axis sediments supporting the importance of hadal microbes in deep-sea nitrogen cycling, via the likelihood of N₂ released into the hadal water column" (see lines 36-38).

We agree that nitrogen cycling has been reported for hadal sites; however, we think that the anammox data provided by Nunoura et al., (2013; 2018) were inconclusive regarding the importance of the process in the hadal realms. We believe that reviewer #1, is familiar with Nunoura et al., (2013) where the authors sampled one sediment core from the bottom axis in Ogasawara trench (9,760 m) and applied qPCR to quantify/amplify ammonia oxidation key genes (*amoA*, *hzsA*, *hao/hzo*), and one denitrification target gene (*nirK*). The authors were successful with cloning sequences of anammox genes (*hzsA* and *hao/hzo*). However, when they performed qPCR on functional nitrogen cycling genes from four different sediment horizons of one sediment core from the Challenger Deep (~10 km depth; Nunoura's et al. 2018), the authors were successful with the *amoA* and *nirK* genes but not with the *hzs*. This caused them to conclude that anammox in CD was insignificant (see also response #2).

In our study we were able to sample a grid of slope and bottom axis sites along the topography of the CD trench. As explained in our response to comment #2, we provide data for both anammox and denitrification, and we believe that our data on anammox contradict Nunoura's suggestion about its lack of importance in Challenger Deep. This contradiction in the findings regarding anammox is reasonable in a sense, considering that in this study we utilized high throughput omics analyses compared to the qPCR approach that provides other important benefits in describing nitrogen cycling.

Regarding the other nitrogen processes reported by Nunoura, we admit that both *amo* and *nirK* were strong clues for ammonia oxidation and denitrification respectively. For denitrification, however, due to the modular nature of the process (many phyla perform different steps of the process), as well as the existence of bacteria that perform truncated denitrification (e.g., Graf et al., 2014, Braker et al., 2010) we believe that more functional denitrification genes are required to demonstrate active denitrification. In our data we identified almost all genes of the denitrification pathway (*narG*, *nirK*, *norB* and *nosZ*) scattered among different phyla. The presence of the last gene of denitrification (*nosZ*) suggests that N₂ is likely released to the abyssal water column. Finally, we believe that the small sampling scale of Nunoura (especially in the Challenger Deep), and the use of three target genes provides information about which processes of nitrogen cycling potentially occur in hadal realms, but certainly does not describe how these nitrogen processes are distributed in the context of the different geochemistries (O₂, NO₃⁻ and NH₄⁺ concentration-wise) among the different CD sites, nor does it describe well the different microbial communities present at the different sites as presented in this study.

For all the above, we believe that our study therefore provides a more complete and more detailed picture of the microbially-mediated CD N cycle.

5. *“Our results ... provide the first data on functional characterization of microbiota in hadal sediments.” Clearly this is incorrect considering the work by Nunoura et al. mentioned above, but also, e.g., in view of recent publications by Peoples and coworkers as well as a host of earlier cultivation-based studies.*

Response: Thank you for pointing this out. We have deleted the sentence.

6. *The dataset described here could be a gold mine, but the value does not come across to the reader. My recommendation is either to expand the paper to a longer format which can be presented in a more specialized journal, or to focus on one or a few aspects which can be treated coherently in proper depth and detail with robust conclusions and clear delineation of the novelty and perspectives for our understanding of the hadal realm. In the present form, the paper covers too many aspects superficially without tying them together, and with the lack of clear research questions, it does not appeal to the broad audience of a generalist journal such as Nature Communications.*

Response: Thank you for the comment. We have revised the manuscript to focus only on 1) the novel microbiome in the CD trench axis; 2) evidence for involvement of CD trench microbiota in the N cycle and 3) evidence for heavy metal biotransformations as potential energy gain/detoxification mechanisms.

Reviewer #2 (Remarks to the Author):

1. Zhou et al. present a study of metagenomes and some metatranscriptomes recovered from sediments within the axis of the Challenger Deep as well as from north or south trench slope.

There are many things to recommend this study. In addition to overcoming all the technical challenges associated with obtaining the sediments the authors have also obtained extensive geochemical data and also have some metatranscriptomes, so that descriptions of activity are also possible. The statistical analyses are robust, the figures are nicely prepared, the bioinformatics methods are well described and in general the materials and methods are excellent, with some exceptions noted below. However, the take home messages from this work do not seem to be particularly surprising. Finding new features for the first time in the microbial populations of the Challenger Deep (CD) is certainly important to specialists in this habitat, but are their discoveries of more general fundamental significance?

Response: Thank you for the positive comment and opinion about general significance. We extensively revised the manuscript and we primarily focus on three aspects of the CD microbiomes as noted above: 1) the novel microbiome in the CD trench axis; 2) evidence for the involvement of CD microbiota in the N cycle and 3) evidence for heavy metal biotransformation as potential energy gain/detoxification mechanisms. We do believe that the new revised manuscript better describes and more clearly addresses the potential microbial activities of the hadal CD, the diversity found at the different sampling sites which is influenced by the distinct geochemical conditions (O_2 , NO_3^- and NH_4^+) that occur along the V-shaped trench. Also, we provide a more complete picture of the CD microbes and their likely contributions to global ocean nutrient cycling. The addition of measurements of heavy metals to our manuscript (total arsenic, arsenate, mercury, total selenium) supports our 'omics' data that indicate active biotransformations for energy/detoxification or even synthesis of more rare amino acids (e.g. selenocysteine) that may support their proteome and their metabolic functions in the hadal realm.

2. Line 61- *This suggests localized evolution as opposed to evolution elsewhere and transport to the CD. This would seem to be refuted by the high 16S similarity among microbes in different benthic hadal and non-hadal settings. Perhaps a hypothesis to evaluate with the MAGs obtained through this study?*

Response: Thank you for this interesting point.

The bottom-axis site (>10km) of Challenger Deep is deepening gradually. This means that a million years ago, the ancestors of microbes in the bottom-axis were subjected to hydrostatic pressure equivalent to that at present day slope sites (6-8km). Such gradual deepening could in theory lead to more isolated communities, and indeed we observe some similarities in composition between CD microbiota and other hadal settings that may share some physico-chemical characteristics. However as noted, we also observed,

hadal communities are not completely distinct, and we found fairly high similarity between CD communities and communities in other non-hadal deep sea sediments. This suggests some degree of connectivity still exists with the upper water column via detritus. To avoid confusion, we have revised the introduction and deleted this statement.

3. Line 114- Any eukarya present? An overview of their diversity could also be briefly described.

Response: Thank you for this insightful suggestion. Indeed, we followed your suggestion and we identified microeukaryotic signatures in our hadal sediment samples. We have added results and discussion starting on line 164. We recovered a total of 3,415 18S miTags from 37 metagenomes that are primarily annotated to Sordariomycetes (Ascomycota) which are known fungal organic matter decomposers with a still unclear role and function in the deep biosphere. Our 33 metagenomes contained fewer than 200 18S miTags. So, we analyzed 18S miTags from three depths 5,400m, 7,143m and ~10,900m to further examine this, and we acknowledge in the text that we may be underestimating/under-sampling the microeukaryotic diversity in CD (see lines 172-174), so our results should be interpreted with caution.

4. Line 124 - The percentages of novel 16S miTags is higher from the CD than from vent and seep sediments and also in bottom-axis versus slope sediments. This is very interesting. What about in comparison to sediments from other water column depths such as abyssal or bathyal?

Response: Thank you for this suggestion. We examined data from 8 sediment samples collected at depths between 1,390 and 3,510m from the South China Sea and Mariana Trench. The average percentage of novel 16S miTags of these non-CD samples is 12%, which is much lower compared to what we report for CD sediment samples (~26%). We now merged these 8 sediment samples with other deep sea sediments samples from hydrothermal vent and seep sediments in Fig.1b. Please see lines starting at 127 and in figure legend 1b.

5. Line 183 and suppl. Fig 5. The metatranscriptomes are not extensively evaluated in this paper. They don't seem to contribute much to the story.

Response: We believe that our current analyses in our revised manuscript of these three metatranscriptomes support our interpretations of the MAG data, and we would like to avoid adding additional stories to this paper, considering that their extensive analyses will make the manuscript too long and less focused. As mentioned, their role was primarily to confirm the functions/relevance of some key pathways indicated by the MAG data. In the revision, we only described data from the most actively transcribed species and affiliated phyla (see lines starting at 253 and Supplementary Fig. 9).

6. Line 203 - Distinct MAGs are described to be present in the slope and bottom-axis, and many others are shared between these two sample types. This is also interesting. But, this study does not take the analyses to the next level and address the kinds of distinct microbes/functions that distinguish these 3 groups?

Response: Thank you for the useful comment. We believe that what causes the distinct distribution of MAGs between slope and bottom sites is the different concentrations of NO_3^- , NH_4^+ and O_2 detected, as described in the “*Geochemistry of Challenger Deep (CD) sediments* section (starting line 94)” and “*Spatial distribution of the microbes in Challenger Deep section* (starting line 208)”. For example, anammoxers are the most highly detected bottom-axis microbes in sediments >12 cmbsf layers where anoxic conditions exist together with high levels of NH_4^+ that sustain anammox bacteria. Please see “*Microbially mediated N_2 release from CD* section (starting at line 328)”. Slope and bottom-axis sediments might also have different organic carbon and nitrogen sources. For microbes surviving in both slope and bottom-axis samples, we took the MAGs a step further and we find they tend to have bigger predicted genome sizes (Fig. 2c) with more carbohydrate-active enzyme (CAZymes) and peptidases genes (Supplementary Fig. 8), which enables them to degrade a wider spectrum of carbohydrates, including detrital proteins than solely slope or bottom-axis distributed microbes. These points are now discussed starting at line 236).

7. Line 220 - the functional characterizations become tedious. This is great stuff for the trench sediment microbiome aficionados, but a slog for others. What is needed is the thirty thousand feet view of the functions present here (or not) in comparison to those present in other marine sediments. Is it really necessary to include descriptions of glycolysis, C3 and C4 metabolite interconversions, and amino acid metabolism in the main body of this paper? If so it should be explained. It does strike me that the pupylation story and its connection to protein turnover in eukarya is valuable.

Response: Thank you for the useful comment. We have moved all these sections now to “*Carbohydrate-Active Enzymes involved in cell wall remodeling and organic matter degradation* Section” in Supplementary Discussion (starting line 51).

8. Line 339. In a similar vein the nitrogen story would benefit from a bigger picture perspective. What is unexpected? Finding anammox in the CD is new, but if it was a surprise then why? Are there aspects of the anammox communities in the CD or their associated geochemical profiles and gene distributions that are distinct from those present in other marine sediments?

Response: Thank you for this useful comment. In our study, we analyzed a significant number of metagenomes representing the different slope and bottom-axis sites to depict a more complete picture of N cycling in hadal zone. The width and depth of CD is 80km and 5km, respectively, in this large water body and in its sediment cores, we did not

find nitrogen fixation genes (see line starting at 372), which suggests any N₂ produced is released and from the sediments to the abyssal water column (see revised Fig.5). In other deep-sea sediments, anammox is occasionally found, but it appears more significant in CD sediments. Due to the funneling effect, CD collects massive amounts of detrital proteins and POM from upper water column and seafloor. This organic N is an important source under these nutrient-poor and high-pressure conditions in CD (see lines 294-311). The important role of anammoxers and denitrifiers is suggested because they would release N₂ from the bottom of the trench to overlying waters, and thus contribute to deep biosphere N cycling (see “*Microbially mediated N₂ release from CD*” section starting at lines 332 and 367). This potential contribution to global N cycling was unexpected (see our conclusion and line starting at 464).

9. Line 384 - Finding genes associated with arsenic detoxification seems novel - is it? Were the arsenic levels (or those of other heavy metals) high?

Response: Indeed, it is novel, and thank you for acknowledging it. In the revised document we now provide measurement of arsenate, total arsenic, mercury and selenium concentrations (starting line 106) to support the idea that heavy metals are biotransformed and detoxified by CD microbes. Total concentrations of the three heavy metals accumulated in the bottom-axis of CD (Supplementary Fig. 1c-e). We also describe potential uses of these compounds in the trench bottom (please see revised section *Crosstalk of arsenic, selenium and sulfur cycling for energy gain and detoxification in CD sediments* starting at line 380). The presence of toxic metals in these hadal sediments explains the presence of detoxification genes in most of our MAGs and also in our three metatranscriptomes (Fig. 3a-c and lines 383-387).

10. Line 399 - what distinguishes figure 6 details from other marine sediments?

Response: Thank you for pointing this out. We now enriched Figure 5 with more details, including the processes of N₂ release to the abyssal water column, heavy metal detoxification and detrital degradation of proteins and organic matter that all appear to occur commonly in CD sediments. What is distinguishable here, is that in our data we show N₂ release compared to other settings (e.g., hydrothermal vents and cold seeps) where there is less or no evidence for N₂ production but evidence for N fixation. We also show the high percentage of genes associated with all these processes (N-cycling, heavy metal detoxification) suggesting their wide use.

11. Line 402. An integrated results and discussion would make it easier for readers to connect the results with their significance.

Response: Thank you for this suggestion. We reorganized the manuscript following reviewer's suggestion.

12. Line 517 - More information on the sample recoveries is needed. The supplementary

data 2 table should describe how samples not collected by lander or submersible were collected - presumably hydrographic cable. Some description of lander pushcore operation is also needed in the supplementary information document or in the main materials and methods. Perhaps the lander is described in another publication? Information on length of time required to bring sediment cores on the ship, the temperature of the recovered cores, and their processing conditions (e.g., room temp or not) would be helpful.

Response: Thank you for this suggestion. The ways (submersible, hydrographic cable or lander) of sample recovery are now provided in the “Sampling method” column of Supplementary Data 2. The lander could return to the surface in 3.5hr with the fastest speed. A picture of the lander with sampler attached is now provided in Supplementary Fig. 2 with details of the sampling. We provide also the *in situ* and recovered temperature of the cores at lines 104 and 482.

13. Line 529 - It is indicated that analysis included arsenic. What else was measured? This additional data should be included.

Response: Thank you for the useful comment. We added the measured concentrations on total arsenic, arsenate, mercury, and total selenium (starting at line 106, Supplementary Fig. 1 c-e and Supplementary data 1)

14. Line 530 - what information is present in the parentheses?

Response: This is Chinese Standard for analysis of marine sediment, GB17378.5-2007 <https://www.chinesestandard.net/PDF/English.aspx/GB17378.5-2007>.

15. Line 539. The metagenome data is described in detail in the paper, but the same is not true for the metatranscriptome data. Its quality and quantity should be described. How can one be confident that the cDNAs generated reflect expression in situ? Perhaps from the types of microbes and genes showing highest expression levels?

Response: Thank you for the suggestion. We provide more information for the

transcriptomics analyses including the amount of sediment samples, concentration and quantification methods and the amount of input total RNA. See lines starting at 534.

Reviewer #3 (Remarks to the Author):

The researchers used sediment collected from 13 slope and axis sites of the Challenger Deep to generate 37 metagenomes from which 586 MAGs were assembled. Metatranscriptome data were also obtained for three samples. The data allowed the authors to survey the potential metabolic capacities and the likely ecological roles of sediment microbes in the Deep. The dataset is of good quality, interesting, and publishable. The manuscript provides new insights into the microbial inhabitants and their potential contributions to biogeochemical cycles (e.g. N, As) and life processes in the hadal biosphere. However, I feel that the manuscript did not make me so excited about the work for the following reasons.

Response: Thank you for the positive comments on our manuscript. We do hope that the revised manuscript will address you concerns.

(1) First, we already know quite a bit about the hadal biosphere, for example, microbial oxidation of hydrocarbons. What are the new discoveries in this study that make it stand out?

Response: Thank you for this comment. We list the novelties in our manuscript below:

1) This is the first study that provides a suite of metagenomic and metatranscriptome datasets that cover the distinct topography of the V-shaped CD. The different sampling sites along the trench have different nutrient concentrations (O₂, nitrate, ammonia), and as we show can sustain different microbial communities. We analyzed and compared our genomic data with abundant reference data from non-CD sediment, CD water column, hydrothermal vents and cold seeps. Also, we detected and quantified the novelty of microbes in the Challenger Deep and compared our findings with other major marine habitats. In addition, we comment the novelty and the different genome sizes as part of adaptations to the ephemeral nutrient sources available in CD.

2) We make clearer now in our manuscript that the genomes of the apparently endemic microbes in CD trench axis sediments encode various carbohydrate active enzymes as well as peptidases that can breakdown organic matter, detrital proteins, or microbial cell walls, that in turn can be utilized as labile carbon sources. In addition, we emphasize the role of specific amino acid pathways for which we have evidence, that can provide ammonia and one-carbon components and support autotrophs and anammoxers in the CD sediments (see lines 296-303). At the reviewers' request, we provide in the revised manuscript measurements for heavy metals (total arsenic, arsenate, total mercury and total selenium) from 10,911 m depth which provide additional support for our genomic and metatranscriptome data that suggest utilization/biotransformation of heavy metals is used as part of energy/detoxification mechanisms (lines 380-441). Also, we provide evidence that CD microbes have higher abundant genes associated with assimilatory

sulfate reduction that can support the synthesis of S-bearing amino acids that upon degradation can provide substrates that can be used in the Krebs cycle (when aerobic respiration is available) and/or support the detoxification of heavy metals (lines 442-451 and Supplementary Fig. 16a).

3) Our data on nitrogen cycling describe nitrification, anammox, and denitrification present in these sediments, as well as coupled with chemolithotrophy as potential feeding modes. Also, our data indicate the potential role of this isolated and endemic (when it comes to microbial communities) hadal realm in N₂ release and in deep-biosphere nitrogen cycling. We also demonstrate anammox genomes with a complete anammox pathway and high abundances of anammoxers in bottom-axis anoxic sediments (> 12cmbsf) of CD (10,900 - 10,911 m) (section starting at line 329). We did not identify genes/microbes related to nitrogen fixation, which indicates that the produced N₂ via the last step of denitrification and anammox (for which we have evidence) is likely released to upper water abyssal zone (lines 372-377).

4) As already mentioned, we extensively revised our manuscript, we highlighted parts that were previously vaguely described to further reflect our novel findings.

(2) Second, I was curious about if the authors have found any evidence for eukaryotic microbes, e.g., fungi in the hadal sediment.

Response: This was an insightful suggestion raised also from Reviewer #2. We followed your suggestion and we extracted 18S miTags from all metagenomes. We identified fungal signatures (Ascomycota; 82% of our 18S miTags) and in particular Sordariomycetes which are known organic matter decomposers with an unclear, yet, role in the deep biosphere. We added text on our manuscript to address the microeukaryotic evidence. See lines starting at 164.

(3) Finally, the hypothesis presented in the manuscript was not exciting and offers no intrinsic aspirations to the reader to further explore the hadal biosphere.

Response: We have extensively revised our initial manuscript and we now present our hypothesis clearly. The differences in O₂, NO₃⁻ and NH₄⁺ concentrations detected at the different sites in this study are now clearly described (see lines starting at 94 and Supplementary Discussion starting at line 28), we revised extensively the section on biotransformation/detoxification of heavy metals, and we added additional data on heavy metals concentrations (total arsenic, arsenate, mercury, total selenium) that we believe provide additional support to our 'omics' data. The high prevalence and expression of *arsM* (methylates arsenite to methylarsenite) in our 'omics' data indicate production of methylated arsenite described as an antibiotic-like compound known to be used by microbes, and also imply potential use of organoarsenicals to confer protection against the low temperatures and high hydrostatic pressures existing in CD (lines 388-406).

(4) One other point is that the authors failed to discuss any ecological or physiological

implications for the discovery of genes encoding for arsenate reduction/detoxification. This can be an important selling point for this work.

Response: Thank you for this comment. We have extensively revised the arsenate reduction/detoxification section to meet this reviewer's expectations and we indicate crosstalk of arsenic, selenium and sulfur cycling for detoxification/energy gain (please see lines starting at 380).

(5) I would also suggest that a comparison and contrast in microbial communities and their metabolism be discussed between the water column and sediment in the Mariana Trench as such data is available in the literature.

Response: Thank you for this comment. We used 16S miTags extracted from Mariana Trench water metagenomes by Gao ZM, et al. (2019) to analyze their microbial communities. We found that the microbial composition in that water column is different from that of CD sediments (please see added Supplementary Fig. 7). We also found more potential hydrocarbon-degrading microbes in CD sediment compared with studies in CD water column (Liu, J., et al. (2019)).

Gao ZM, et al. In situ meta-omic insights into the community compositions and ecological roles of hadal microbes in the Mariana Trench. *Environ Microbiol* 21, 4092-4108 (2019)

Liu, J., et al. Proliferation of hydrocarbon-degrading microbes at the bottom of the Mariana Trench. *Microbiome* 7, 47 (2019).

(6) The statement line 418-423 is confusing. Genome size for MAGs unique to the slope or axis is smaller than those shared between slope and axis? How does this offer an advantage to microbial adaptation in the trench?

Response: We agree with the reviewer that our statement was confusing. It is rephrased now as follows: “The average genome size of the ubiquitous CD-distributed MAGs was ~16% larger when compared to the MAGs unique to the slope sites, and ~11% larger when compared to the unique bottom-axis MAGs (Fig. 2c)” (see lines 230-232). We believe that what causes the distinct distribution of MAG sizes between slopes and bottom sites is the different concentrations of NO_3^- , NH_4^+ and O_2 detected at the bottom and slope sites, as described in the “*Geochemistry of Challenger Deep (CD) sediments*” section (starting line 94), the “*Microbially mediated N_2 release from CD*” (starting line 328) as well as in the Supplementary Discussion (starting at line 119). We also suggest that reduced genome size could be an adaptation that lowers the metabolic cost associated with microbial DNA replication, potentially providing increased fitness under nutrient limiting conditions. Bacterial genomes tend to increase in size by aggregating adaptive gene modules that can provide greater metabolic flexibility. Metabolic flexibility might be an advantage for CD microbes inhabiting both slope and bottom-axis sites where different (and likely ephemeral) available energy pools exist, and this justifies the 11%-16% larger genome size compared to the MAGs that are unique to the slope and axis samples (please start at line 230). For microbes surviving in both slope and bottom-axis samples, they tend to have bigger genome size (Fig. 2c) with more carbohydrate-active enzyme (CAZymes) and peptidase genes (please see added Supplementary Fig. 8), which enable them to degrade a wider spectrum of carbohydrates, including detrital proteins than solely slope or bottom-axis distributed microbes. These points are now discussed in lines in this same section).

(7) *There are a number of syntax errors in the manuscript, e.g., line 415-417.*

Response: Corrected following the reviewer’s suggestion.

REVIEWER COMMENTS

Reviewer #1 (Remarks to the Author):

As noted in my review of the previous version of this manuscript, the data is of high quality and represents a gold mine, which could support a whole series of papers. My recommendation was to either expand the paper to a long format and send it to another journal that accepts such a format, or to focus it to one or a few aspects of particular novelty – leaving other aspects to other publications. The authors have chosen not to follow any of these suggestions but have instead kept the diverse and somewhat superficial and descriptive presentation while fixing some specific issues. Thus, the paper still lacks focus as well as research questions and/or hypotheses, and the broader significance of many of the findings remains unclear. The strongest part of the paper is the global analysis of the metagenomes, which is clearly novel for the hadal realm and could, with some expansion of the discussion, carry the entire paper. Other major conclusions seem either unsurprising or overstated. Citing three consecutive sections from the Abstract:

1) “Functional analyses of the retrieved MAGs suggest that these hadal microbes are mostly heterotrophic degraders that utilize detrital biopolymers as carbon sources.” (l. 31-32) This is a robust conclusion, but isn't it rather obvious that this must be the case based on previous biogeochemical and microbiological studies of trench axis sediments? Was there really a question about this? What else would drive energy flow and thereby support the communities? Nothing in the Introduction hints at the necessity of exploring this question.

2) “The capacity of the microbes to reduce arsenate and selenate as a detoxification and/or energy gain strategy is supported by the presence and the enriched transcription of genes related to arsenic biotransformations in our MAGs and metatranscriptomes, respectively.” (l. 32-36). These findings are interesting, but their significance cannot be evaluated due to the lack of comparison to other relevant environments. Do the hadal sediments stand out relative to other marine sediments in general? And if so, does the differences correlate with As/Se abundances? The comparison to Tara Ocean water column data is irrelevant considering the differences in biogeochemistry which leaves only a comparison to the Guaymas Basin, but this lacks biogeochemical perspective.

3) “We report MAGs from anammox bacteria dominant in CD bottom-axis sediments supporting the importance of hadal microbes in deep-sea nitrogen cycling, via the likelihood of N₂ released into the abyssal water column.” (l. 36- 38) As mentioned in my previous review and now acknowledged in the manuscript, previous studies have indicated the presence of anammox bacteria in hadal sediments. I miss a clear explanation of how the new data represent a step change (and not just an incremental one) in our understanding of benthic nitrogen cycling. I understand that documentation of the anammox pathway in the MAGs strengthens the case, but it is not permissible to draw conclusions about the importance of the process in the N cycle as the authors do here and in the conclusion (“The potential N₂ yield via anammox or denitrification can imply an essential contribution from the hadal sediments to the global nitrogen cycle” l. 466-8). Considering the very small area covered by hadal trench axis sediments worldwide, it seems highly unlikely that their contribution would be “essential”, and there is no discussion of how and why this would be the case.

In addition to the aspects mentioned above, the new version now includes a section on eukaryotes, which is neither covered in the abstract nor justified in the introduction, and which seems inconclusive due to unexplained “under-sampling of microeukaryotic diversity in CD sediments.” (l. 166-7). Like other parts of the manuscript, this is either too little or too much.

Specific comments:

76: “Vertical exchange” – the slope angle of hadal trenches is quite small, so the horizontal distance is much larger than the vertical one.

80: 16S rRNA gene studies, I believe.

99: “similar and constant” is ambiguous – please rephrase.

136: Does the hadal realm really qualify as an extreme environment in this respect? Is there any evidence that high hydrostatic pressure is a strong selective force?

189: “lineages” at which taxonomic level? Genera?

233-236: My impression is that hadal trench sediments are relatively rich in substrates compared to the surrounding abyssal plains. What supports the claim for “nutrient the limiting conditions” here? (note also the typo).

300-301: CO₂ is not an energy source for CO₂ fixers.

304-311: What are the perspectives of these findings?

349-351: How do the findings “agree” with those from the SCS?

Reviewer #2 (Remarks to the Author):

The manuscript is greatly improved as a result of the additional analyses provided and the tighter focus.

I still do not see data on the temperature of the recovered cores.

Line 375. Has N₂ fixation been reported in abyssal waters? This should be cited. If it has not been found then there is no need to suggest where the released N₂ diffuses (or is vertically transported by other means) except that it must go elsewhere in the water column.

Line 399. Cryoprotection refers to protection from freezing which cannot happen at hadal temperatures and pressures.

Reviewer #3 (Remarks to the Author):

The authors have adequately addressed reviewers' comments, especially did analysis of As species in the sediment to support their arguments on arsenate-arsenite transformation. Perhaps the abstract can be written a little better to present the major findings of this work and take it to a higher level.

REVIEWER COMMENTS

Reviewer #1 (Remarks to the Author):

Q1: *As noted in my review of the previous version of this manuscript, the data is of high quality and represents a gold mine, which could support a whole series of papers. My recommendation was to either expand the paper to a long format and send it to another journal that accepts such a format, or to focus it to one or a few aspects of particular novelty – leaving other aspects to other publications. The authors have chosen not to follow any of these suggestions but have instead kept the diverse and somewhat superficial and descriptive presentation while fixing some specific issues. Thus, the paper still lacks focus as well as research questions and/or hypotheses, and the broader significance of many of the findings remains unclear. The strongest part of the paper is the global analysis of the metagenomes, which is clearly novel for the hadal realm and could, with some expansion of the discussion, carry the entire paper. Other major conclusions seem either unsurprising or overstated. Citing three consecutive sections from the Abstract:*

R1: We appreciate this reviewer's concern about the breadth of the paper. We have not intentionally ignored this reviewer's request to focus on fewer topics, but find his/her request in direct contradiction to the requests of our other two reviewers, who asked us to in fact expand on our topics by adding discussion of any available data on microbial eukaryotes. We believe that this first metagenome-enabled comparison of trench bottom-axis vs. slope sediment microbial communities of Challenger Deep should provide a broad overview of what is observed, which we do provide, but to avoid being "superficial" it should dig into a few themes of interest, and we have done this for the themes reported in the paper. Our other reviewers appreciated this, and did not believe that these discussions were superficial or too broad for this study. In deference to this reviewer, we have moved several themes to the supplementary and we have refocused our discussion on comparisons of the two Challenger Deep habitats (slope vs. bottom-axis), and where possible, comparisons with other trench and deep-sea sedimentary settings in the region.

We also thank this reviewer for pointing out the inadequately conveyed description of our primary research questions that drove this study. We have now included in our introduction the following statements (lines 78-82): *"These data together with metatranscriptome data for bottom-axis sites are used to determine if the inferred metabolic capacity and ecological roles of microorganisms found along the slope and bottom-axis sediments show distinctions, and whether they differ from what is observed in studies of abyssal and other hadal deep-sea sediments in the region",* and *"However, detailed genomic analysis is required to provide a complete overview of N₂ cycling in the hadal zone"* (lines 55-56), that we clarify the primary research questions of this study.

In addition, we agree with this reviewer that the context/significance of each major finding reported was inadequate because we did not indicate whether or not any of the activities/metabolic capabilities reported are typical of deep-sea microbial communities. We

edited accordingly each topic discussed in this paper to include this information (e.g., sections on *Recycling of detrital organic matter and utilization of hydrocarbons, CO₂ fixation, Crosstalk of arsenic, selenium and sulfur cycling for energy gain and detoxification in CD sediments*; lines 257-258, 304-306, 330-332, 362-365, 382-400 and 421-423) to satisfy this reviewer's request. Also, we provide examples from geographically "nearby" deep-sea locations, including the South China Sea, (please see lines: 123-124, 306-307, 333-334); however, we need to make clear that this study aims to describe activities/metabolic capabilities at different sites/habitats along the CD rather than to review the metabolic potential of typical marine deep-sea sediments vs. CD.

We understand the reviewer's perspective that most of our findings are "unsurprising"; however we here describe only a tiny fraction of our metagenomic dataset that was successfully annotated (functionally and taxonomically). We point out that this is due to the fact that many/most of the genes encoded in the reconstructed MAGs have no homologues in public databases (please see lines 242-245). A similar restriction is seen in the handful of metatranscriptomes that we analyzed in this study that show > 82,000 actively transcribed genes of unknown function which cannot be further extrapolated to describe potential metabolic activities that might be unique to this hadal realm. As this reviewer acknowledges the data that we do present are "*clearly novel for the hadal realm*" and I think we all agree that these datasets will be a goldmine of information for anyone interested to dig further into microbial genomic potential in the future as databases and annotations improve (lines 248-252).

Also, while heterotrophic activities are of course expected in Challenger Deep we think we make an important contribution to our understanding of the hadal realm by providing a) genetic evidence of what types of heterotrophic processes might occur in CD (e.g., utilization of carbohydrates, hydrocarbons, aromatic organics, proteinaceous compounds, recycling of amino acids) and b) how these metabolic capacities are shaped between bottom-axis and slope communities. Further, we believe that we now adequately highlight these contributions in our manuscript.

Q2: *"Functional analyses of the retrieved MAGs suggest that these hadal microbes are mostly heterotrophic degraders that utilize detrital biopolymers as carbon sources." (l. 31-32) This is a robust conclusion, but isn't it rather obvious that this must be the case based on previous biogeochemical and microbiological studies of trench axis sediments? Was there really a question about this? What else would drive energy flow and thereby support the communities? Nothing in the Introduction hints at the necessity of exploring this question.*

R2: As we mentioned above, we agree with the reviewer that heterotrophy would logically dominate in deep/abyssal/hadal sediments and thus we corrected our text accordingly to avoid any implication that this would be a surprise (please see introduction lines 49-50, and also discussion *Recycling of detrital organic matter and utilization of hydrocarbons* lines starting from 254). However, we would like to point out that the literature regarding how this heterotrophy (energy sources and mechanisms wise) is performed in trenches is still limited.

Besides discussion of genetic evidence for potential heterotrophic processes that may occur along the CD, we also provide information on the types of enzymes encoded in our MAGs that can facilitate the breakdown of organic matter and macromolecules. Regarding the CAZymes that we report, we provide information on the specificity of the most dominant enzyme families encoded in the MAGs, and how their function can be interpreted to Challenger Deep. All this information exists in the supplementary text that we submitted, and we have no objection to moving this information into the main text if desired.

Q3) “The capacity of the microbes to reduce arsenate and selenate as a detoxification and/or energy gain strategy is supported by the presence and the enriched transcription of genes related to arsenic biotransformations in our MAGs and metatranscriptomes, respectively.” (l. 32-36). These findings are interesting, but their significance cannot be evaluated due to the lack of comparison to other relevant environments.

R3: We would like to thank the reviewer for finding the arsenate/selenate story interesting. We rephrased our introduction to reflect this reviewer’s point (please see lines 62-66). We believe that the absence of data on arsenic cycling from other hadal or deep-sea sediments, which precludes any direct comparisons, does not make the arsenic/selenate stories from CD less significant. Further, active transcription of genes related to utilization of arsenate/selenate in hadal trenches has never been reported previously, and we believe that the genetic evidence that we provide shows that As/Se biotransformations have the potential to be important available energy pools in CD. We believe that Challenger Deep is a remote ecosystem that can retain types of “ancient metabolisms” for energy gain. There is extensive literature that suggests arsenic cycling (As(III)/As(V) oxidation/reduction by microbes) could be active in remote and/or energy challenged non-hadal or deep-sea environments (e.g., Sforza et al., 2014, Chen et al., 2020; Wells et al., 2020), and here we make this argument for CD. Our data show that CD is a setting where energy acquired via biotransformations of arsenic species is feasible, while the enzymes we report are described as “enzymes of early origin” (arsenate reductases, arsenite oxidases) and commonly used as evolution markers in paleogeochemical studies (e.g., Duval et al., 2008).

Our efforts to find arsenic/selenate concentrations and/or genetic data from deep sea/abyssal/hadal sediments were unsuccessful with regards to selenate. Regarding arsenic, *acr3* and *arsB* genes were recovered from seawater metagenomes from the hadal Yap trench; Zhang et al., 2018 (please see lines 383-384). Even the comparison that we provide with data acquired from deep sediments in Guaymas Basin is difficult to interpret due to the extremely reductive and sulfidic environment existing in Guaymas Basin that can trigger thermodynamically favorable abiotic As transformations (lines 397-400). Masuda et al., (2019) results on the vertical profile of As species (IODP 338 subseafloor expedition Nankai Trough; 0-1,200 mbsf) clearly indicate that As becomes mobilized in deep sea sediments from intense microbial activity, which could support the significance of As cycling as a bioenergetic pathway in the energy-limited setting of CD. However, Masuda’s study lacks genetic evidence for microbial As mobilization in the subseafloor of Nankai Trough, and thus, we cite their work in our manuscript (lines 385-386), but we cannot make any direct comparisons.

References:

Sforna, M., Philippot, P., Somogyi, A. et al. Evidence for arsenic metabolism and cycling by microorganisms 2.7 billion years ago. *Nature Geosci* 7, 811–815 (2014)

Chen P, Zhang HM, Yao BM, Chen SC, Sun GX, Zhu YG. Bioavailable arsenic and amorphous iron oxides provide reliable predictions for arsenic transfer in soil-wheat system. *J Hazard Mater* 383, (2020).

Wells, M., Kanmanii, N.J., Al Zadjali, A.M. et al. Methane, arsenic, selenium and the origins of the DMSO reductase family. *Sci Rep* 10, 10946 (2020).

Duval S, Ducluzeau AL, Nitschke W, Schoepp-Cothenet B. Enzyme phylogenies as markers for the oxidation state of the environment: The case of respiratory arsenate reductase and related enzymes. *BMC Evol Biol* 8, (2008).

Zhang X, Xu W, Liu Y, Cai M, Luo Z, Li M. Metagenomics reveals microbial diversity and metabolic potentials of seawater and surface sediment from a hadal biosphere at the Yap Trench. *Front Microbiol* 9, 2402 (2018)

Masuda H, Yoshinishi H, Fuchida S, Toki T, Even E. Vertical profiles of arsenic and arsenic species transformations in deep-sea sediment, Nankai Trough, offshore Japan. *Prog Earth Planet Sci* 6, 28 (2019)

Q4: *Do the hadal sediments stand out relative to other marine sediments in general? And if so, does the differences correlate with As/Se abundances?*

R4: We do believe that a *general* comparison to typical marine sediments worldwide would be useful in a review paper or in a paper focused on detection of, and microbial transformations of metals in hadal trenches. These topics are outside the scope of this paper. Also, comparisons with other marine sediments might not be relevant here, considering that marine shallow sediments and/or coastal sediments receive extensive As pollution due to human activities (e.g., Neff, 1997). Here, we provide the As/Se story in the context of describing our findings along the CD trench (slope vs. bottom-axis) and we also compare them with our available data from non-hadal sites to make our arguments (Supplementary Figure 1c). We believe our findings will help future investigations of biogeochemical cycling of metals in hadal trenches.

Reference :

Neff JM. Ecotoxicology of arsenic in the marine environment. *Environ Toxicol Chem* 16, 917-927 (1997).

Q5: *The comparison to Tara Ocean water column data is irrelevant considering the differences in biogeochemistry which leaves only a comparison to the Guaymas Basin, but this lacks biogeochemical perspective.*

R5: The reviewer is correct. The Tara Ocean comparison is now removed. Regarding the comparison to Guaymas Basin we have already replied and we added text to address this reviewer (please see lines 395-400).

Q6: *“We report MAGs from anammox bacteria dominant in CD bottom-axis sediments supporting the importance of hadal microbes in deep-sea nitrogen cycling, via the likelihood of N₂ released into the abyssal water column.” (l. 36- 38) As mentioned in my previous review and now acknowledged in the manuscript, previous studies have indicated the presence of anammox bacteria in hadal sediments. I miss a clear explanation of how the new data represent a step change (and not just an incremental one) in our understanding of benthic nitrogen cycling. I understand that documentation of the anammox pathway in the MAGs strengthens the case, but it is not permissible to draw conclusions about the importance of the process in the N cycle as the authors do here and in the conclusion (“The potential N₂ yield via anammox or denitrification can imply an essential contribution from the hadal sediments to the global nitrogen cycle” l. 466-8). Considering the very small area covered by hadal trench axis sediments worldwide, it seems highly unlikely that their contribution would be “essential”, and there is no discussion of how and why this would be the case.*

R6: We respectfully disagree with the reviewer. Our data on anammox in CD are in full accordance with the new and fascinating findings of Thamdrup et al., 2021 that not only reports anammoxers, but also hadal trenches as hot spots of N₂ loss due to fully functional anammox activity under such elevated hydrostatic pressures. We were clear from our initial submission that anammox is a key process in hadal sediments, and we provided an in-depth analysis of the MAGs that encoded the complete anammox pathway and also a phylogeny of the key anammox enzymes found in our MAGs vs. the anammox enzymes publicly available (please see lines 335-348). We also reported in our initial submission that MAGs annotated to anammoxers were retrieved only from the deeper bottom-axis sediments (e.g. bottom-axis sample T3L11 18-21cbsf) and not the slope sites (see lines 324-327), and thus we argued for heterogeneous N₂ loss along the CD sediments (slope vs bottom-axis) (see lines 365-369). Indeed, the trenches cover a small area worldwide, however, the intensity of anammox and the amount of N₂ produced in the trenches also varies. The incubations of Thamdrup et al., 2021 report % of N₂ production by anammox between 67±13% up to 98% for the Atacama and Kermadec Trench, respectively (diffusive fluxes; mmol N m⁻² d⁻¹). These numbers exclude the N₂ production from denitrification, that although seems to contribute less than anammox in the trenches, can only add to the total N₂ loss from the hadal biosphere. Unfortunately, similar incubation experiments are absent for Challenger Deep which could provide robust indications on the N₂ loss from CD by anammox/denitrification, and how these are compared to other similar hadal settings.

Also, we acknowledge that research published by Nunoura was pivotal by reporting anammox/denitrification in hadal sediments, however, describing bacterial lineages and genes involved in N₂ loss processes, as well as their distribution along the different sites of CD (slope vs bottom-axis), as we have done here, is equally important. We mentioned in our previous response letter that processes involved in N₂ loss (e.g., denitrification) can be modular and that not all bacterial lineages can perform all steps of these processes (truncated vs. complete denitrification; Dalsgaard et al., 2014 and Graf et al., 2014). This underlines the necessity for a more complete description of N₂ loss pathways (including denitrification) if we want to address deep sea nitrogen cycling and potential N₂ loss. It is well known that denitrification genes (*narG*, *napA/B*, *nirK/S* *norB/C*, *nosZ*) are scattered between denitrifying phyla, and although we had evidence of commonly used key denitrification genes in our data (e.g., *narG* and *nirK*), we also identified the genes (*norBC* and *nosZ*) that actually produce and release N₂ in the final steps of the process. We believe that this is more robust evidence of complete denitrification in CD sediments, and this was not previously reported. Additionally, how processes involved in N₂ loss are distributed along the hadal trench, how similar, or not, are the genes for enzymes involved in these processes with those in public databases, are questions that need to be answered if we want to be precise about describing N cycling in deep sediments. This is a more complete description/proof of the genes for the process than reporting individual key genes. We rephrased sentences in our text to focus on these points (please see lines 327-329, 360-373 and Supplementary Discussion lines 160-163).

Finally, we feel that there are many open questions that remain about nitrogen cycling in the deep sea and, and in particular in hadal settings. We sincerely hope that this revised manuscript now addresses all these concerns about the importance of hadal realms in N₂ loss, and N cycling.

Reference:

Thamdrup B, et al. Anammox bacteria drive fixed nitrogen loss in hadal trench sediments. Proc Natl Acad Sci USA 118, e2104529118 (2021).

Dalsgaard T, et al. Oxygen at nanomolar levels reversibly suppresses process rates and gene expression in anammox and denitrification in the oxygen minimum zone off Northern Chile. mBio 5, e01966-01914 (2014)

Graf DR, Jones CM, Hallin S. Intergenomic comparisons highlight modularity of the denitrification pathway and underpin the importance of community structure for N₂O emissions. Plos One 9, e114118 (2014).

Q7: *In addition to the aspects mentioned above, the new version now includes a section on eukaryotes, which is neither covered in the abstract nor justified in the introduction, and which seems inconclusive due to unexplained “under-sampling of microeukaryotic diversity in CD sediments.” (l. 166-7). Like other parts of the manuscript, this is either too little or too much.*

R7: The addition of discussion of metagenomic data from eukaryotes to the manuscript was requested from reviewers #2 and #3. Reporting or discussing eukaryotic signatures from hadal sediments was not among the original scientific questions that we were aiming to address. Upon request from reviewers #2 and #3 we performed additional analysis of our data and we conveyed our findings as well as discussion of potential caveats about interpretations of presence/absence of particular genes or taxonomies in a study that did not originally target eukaryotic data. These additions satisfied reviewers #2 and #3.

Specific comments:

Q8: 76: *"Vertical exchange" – the slope angle of hadal trenches is quite small, so the horizontal distance is much larger than the vertical one.*

R8: Yes, we have deleted it.

Q9: 80: *16S rRNA gene studies, I believe.*

R9: Corrected. Please see line 72.

Q10 99: *"similar and constant" is ambiguous – please rephrase.*

R10: Corrected. Please see line 93.

Q11: 136: *Does the hadal realm really qualify as an extreme environment in this respect? Is there any evidence that high hydrostatic pressure is a strong selective force?*

R11: That is a very interesting comment and we thank the reviewer for that. High hydrostatic pressure (> 10MPa) is considered to be a strong selective force in the deep ocean and is often compared to the selective effect that energy sources have on shaping the microbial community structure of the deep ocean (Xiao et al., 2021). We strongly believe that the effect of pressure depends on the phyla described and the processes performed. As an example, the anammoxers identified in Atacama and Kermadec trenches seem to be adapted to the elevated hydrostatic pressures that exist in the trenches rather than to be diversified into new phyla due to action of hydrostatic pressure as a selective force. However, literature suggests that elevated pressures have profound effects on intracellular processes including gene transcription, expression, and protein translation (e.g., piezo-related expression of glutamine; e.g., Ikegami, et al., 2000) as well as in protein evolution in terms of structural stability, functional mechanisms of enzymes, and membrane composition (e.g., Somero, 1990). We do believe that the elevated pressures that exist in CD have the potential to affect the above-mentioned intracellular processes as well as to shape the sediment microbial communities of CD at protein/enzyme/lipid level. The role of elevated hydrostatic pressures in CD would be more adequately explored using pressure-controlled in situ experiments that will involve transcriptomic as well as proteomic profiling.

References:

Xiao X, Zhang Y, Wang FP. Hydrostatic pressure is the universal key driver of microbial evolution in the deep ocean and beyond. *Env Microbiol Rep* 13, 68-72 (2021).

Ikegami A, et al. Glutamine synthetase gene expression at elevated hydrostatic pressure in a deep-sea piezophilic *Shewanella violacea*. *FEMS Microbiology Letters* 192, 91-95 (2000).

Somero GN. Life at low volume change : hydrostatic pressure as a selective factor in the aquatic environment. *Am Zool* 30, 123-135 (1990).

Q12: 189: *“lineages” at which taxonomic level? Genera?*

R12: We deleted the context on MAG novelty, to avoid misinterpretation of our data.

Q13: 233-236: *My impression is that hadal trench sediments are relatively rich in substrates compared to the surrounding abyssal plains. What supports the claim for “nutrient the limiting conditions” here? (note also the typo).*

R13: We want to thank the reviewer for this comment. Indeed, the total organic carbon (TOC) measured in bottom-axis sites of the CD trench is often higher than the TOC from the surrounding abyssal plains. However, CD has minimal terrestrial inputs as it is situated away from the continent. This results in TOC being ~ 0.2%-0.6% in CD sediments (Glud et al. 2013; Luo et al. 2018; Hiraoka et al. 2020), which is up to five times lower compared to the TOC reported in other trenches (e.g., Izu-Bonin, Tonga Trench sediments TOC >1%) (Luo et al. 2018). The lower TOC measured in CD sediments places the trench among oligotrophic environments.

References:

Luo, M., Glud, R. N. et al. Benthic carbon mineralization in hadal trenches: Insights from in situ determination of benthic oxygen consumption. *Geophysical Research Letters*, 45, 2752–2760 (2018).

Glud RN, et al. High rates of microbial carbon turnover in sediments in the deepest oceanic trench on Earth. *Nat. Geosci.* 6, 284-288 (2013).

Hiraoka, S., Hirai, M., Matsui, Y. et al. Microbial community and geochemical analyses of trans-trench sediments for understanding the roles of hadal environments. *ISME J* 14, 740–756 (2020).

Q14: 300-301: *CO₂ is not an energy source for CO₂ fixers.*

R14: Corrected please see lines 273-274

Q15: 304-311: *What are the perspectives of these findings?*

R15: The Pup-proteasome pathway in bacteria flags proteins for degradation in a similar way like ubiquitination does in eukaryotes. Experimental studies showed that this pathway is highly expressed in bacteria under nitrogen limiting conditions (Elhar et al., 2014), and thus it is

suggested that Pup-driven proteasomal degradation recycles amino acids in bacteria and provides nitrogenous precursors used in central metabolism (Muller and Weber-Ban 2019). We have evidence that genes of the Pup-proteasome pathway are encoded in our MAGs (AOAs and NOBs) and we suggest that this might be an adaptation that can sustain microbial metabolism in CD, where the availability of nutrients changes with sediment depth and with site. We have moved this text to the supplementary discussion where we explain in more detail about this pathway (see Supplementary Discussion lines 75-86).

References:

Elharar, et al. Survival of mycobacteria depends on proteasome-mediated amino acid recycling under nutrient limitation. *EMBO Journal* 33.16(2014):1802-1814.

Muller AU, Weber-Ban E. The bacterial proteasome at the core of diverse degradation pathways. *Front Mol Biosci* 6, (2019) <https://doi.org/10.3389/fmolb.2019.00023>.

Q16: 349-351: *How do the findings “agree” with those from the SCS?*

R16: Our findings agree with the detection of ‘*Ca. Scalindua*’ in the Ogasawara Trench (9,760 m) and the high abundance of anammoxers in deep sediments from South China Sea (2,610 m), identified with hzs and 16S rRNA gene cloning, respectively. This is now pointed out in lines 330-334.

Reviewer #2 (Remarks to the Author):

Q1: *The manuscript is greatly improved as a result of the additional analyses provided and the tighter focus.*

R1: We thank the reviewer for this positive comment.

Q2: *I still do not see data on the temperature of the recovered cores.*

R2: Unfortunately during our sampling in CD we experienced technical problems with our temperature sensors so we were not able to record the in situ temperature of the sediment cores. However, the temperature sensor was functioning on our lander and we were able to record the temperature ~50 cm above the sediment surface. This information is now added to Supplementary data 2 (please see “bottom-water temperature”). The bottom-axis sites are located at a flat plain covered with very thick sediments, so the temperature along the sediment column (at least up to 21 cmbsf that we sampled) should be the same as that of the overlying water.

Q3: Line 375. Has N₂ fixation been reported in abyssal waters? This should be cited. If it has not been found then there is no need to suggest where the released N₂ diffuses (or is vertically transported by other means) except that it must go elsewhere in the water column.

R3: We agree with this reviewer on this point and we have deleted this statement (please see line 362). N₂ fixation has a high energy cost (~800 kJ per N₂ molecule reduced; Karl et al. 2002), and it is mainly reported in the euphotic environment. However, deep-sea N₂ fixation has been reported in organic-rich environments that have enhanced productivity like methane seeps, mud volcanoes and hydrothermal vents (Dekas et al. 2018). Regarding the hadal realms, nitrogen fixation genes have not been detected either in CD water column (Gao et al. 2019) or sediment metagenomes (this study), and thus we speculate that this process is unlikely to occur in the oligotrophic CD environment. This is now noted on lines 362-373.

References:

Karl D, et al. Dinitrogen fixation in the world's oceans. *Biogeochemistry* 57, 47–98 (2002).

Dekas AE, et al. Widespread nitrogen fixation in sediments from diverse deep-sea sites of elevated carbon loading. *Environ Microbiol* **20**, 4281-4296 (2018).

Gao ZM, et al. In situ meta-omic insights into the community compositions and ecological roles of hadal microbes in the Mariana Trench. *Environ Microbiol* **21**, 4092-4108 (2019).

Q4: Line 399. Cryoprotection refers to protection from freezing which cannot happen at hadal temperatures and pressures.

R4: We thank the reviewer for this comment. We rephrased accordingly see lines 403-404.

Reviewer #3 (Remarks to the Author):

Q1: The authors have adequately addressed reviewers' comments, especially did analysis of *As* species in the sediment to support their arguments on arsenate-arsenite transformation. Perhaps the abstract can be written a little better to present the major findings of this work and take it to a higher level.

R1: We thank the reviewer and we appreciate the positive comment. We rephrased the abstract to fulfill also reviewer's #1 request (please see lines 18-32).

REVIEWERS' COMMENTS

Reviewer #1 (Remarks to the Author):

The authors have responded in detail to my comments to the previous version and made further edits to the manuscript. While the changes are not extensive, the result is a much more streamlined and balanced presentation with a broader appeal. I have only a few additional comments.

The closing sentence in the paragraph on the high number of novel organisms in the trench axis sediments (l. 130-32; "Microbial communities surviving in extreme environments are thought to evolve faster compared to microbial communities inhabiting non-extreme environments" [note mistake in "environments"]) needs either qualification or moderation. If the authors imply that this is the driver of the degree of novelty(?) other explanations should be evaluated and discarded. An obvious alternative would be that the trench axis samples include anoxic communities, which are underrepresented in the existing databases. Indeed, Fig. S4 suggests that the trend in novelty score is driven by the deeper, anoxic sediment layers. We must assume that anoxia is a strong selective force, and hence that community composition changes substantially along the geochemical gradients in the sediment.

l. 204-5 now reads as if the new results differ from previous results from CD sediments, but Fig. S7 only indicates previous results from the water column. If there are previous results from the sediments, this should be highlighted. Moreover, it looks to me like at least some trench axis sites plot together with slope sites?

l. 255: "different rates" relative to what? Different along the trench axis, with depth in the trench, or what?

l. 19, 32, 73, 80, 264, 328, 367, 368, 465, 470: "along" is used repeatedly in cases where I think "across" would be more appropriate. Along would be appropriate, e.g., for trends along the trench axis, but as understand it, it is used for trends across slope and axis sites.

Suppl. l. 150-8: This is copy-paste repetition of part of the previous paragraph.

REVIWER COMMENTS

Reviewer #1 (Remarks to the Author):

Q1. *The authors have responded in detail to my comments to the previous version and made further edits to the manuscript. While the changes are not extensive, the result is a much more streamlined and balanced presentation with a broader appeal. I have only a few additional comments.*

R1: We thank reviewer #1 for the insightful comments/suggestions that helped us to achieve a more balanced presentation of our study. Line numbers refer to tracked changes version.

Q2. *The closing sentence in the paragraph on the high number of novel organisms in the trench axis sediments (l. 130-32; "Microbial communities surviving in extreme environments are thought to evolve faster compared to microbial communities inhabiting non-extreme environments" [note mistake in "environments"]) needs either qualification or moderation. If the authors imply that this is the driver of the degree of novelty(?) other explanations should be evaluated and discarded. An obvious alternative would be that the trench axis samples include anoxic communities, which are underrepresented in the existing databases. Indeed, Fig. S4 suggests that the trend in novelty score is driven by the deeper, anoxic sediment layers. We must assume that anoxia is a strong selective force, and hence that community composition changes substantially along the geochemical gradients in the sediment.*

R2. We agree with the reviewer, and we rephrased as follows (see lines 143-147): “Anaerobic microbial communities such as those detected in bottom-axis sediments are underrepresented in existing databases. Supplementary Fig. 4 indicates that increasing novelty scores in bottom-axis sediments are primarily driven by sediment depth and anoxia. Nutrient availability, which is often lower with depth and anoxia are selective forces known to structure microbial communities in deep-sea sediments³¹.”

Varliero G, Bienhold C, Schmid F, Boetius A, Molari M. Microbial diversity and connectivity in deep-sea sediments of the South Atlantic polar front. *Front Microbiol* **10**, (2019).

Q3. *l. 204-5 now reads as if the new results differ from previous results from CD sediments, but Fig. S7 only indicates previous results from the water column. If there are previous results from the sediments, this should be highlighted. Moreover, it looks to me like at least some trench axis sites plot together with slope sites?*

R3. The reviewer is right, and we now clarify this to avoid any misinterpretations. We performed two different PCoA analyses; one was to convey differences in the microbial community composition between slope and bottom-axis (main text Fig. 2b) and the other PCoA analysis was to identify overall differences in the microbial composition between CD sediments (our study) and CD water column (data from Gao et al., 2019) (Fig. S7). The PCoA analysis in Fig 2b (slope vs. bottom-axis) is created using the relative abundance of MAGs, while the PCoA analysis in Fig S7 (sediments vs. water column) using the relative abundance of 16S miTag data. Fig S7 shows that the 16S miTag data from the water column do not overlap with those from the sediments, which is what we wanted to convey. However, the degree of separation between water column and sediment 16S miTag data is large enough to overwhelm any differences

between slope and bottom-axis communities. For this reason, we created a new Fig S7 that contains a panel **b** in which the 16S water column data are excluded (please see below). As you can see in Fig S7b the same clear separation using the 16S miTag data exists between slope and bottom-axis as that reported using the MAG data (main text Fig 2b).

We also rephrased our text to further clarify. Now it reads (please see lines 187-193): “*Principal coordinate analysis (PCoA) using the relative abundance of MAGs and the Bray-Curtis dissimilarity index confirmed a discrete separation of microbial communities between slope and bottom-axis sediments (Fig. 2b). This spatial separation was more distinct compared to what was observed previously using 16S rRNA data²¹. In our study, we also observed a community composition in CD sediments that is distinct in comparison to previously reported communities in the CD water column²⁹ (Supplementary Fig. 7).*”

Q4. l. 255: “different rates” relative to what? Different along the trench axis, with depth in the trench, or what?

R4. We have corrected the sentence (see lines 239-241). Now it reads: “Due to the funneling effect, the V-shaped CD accumulate organic debris at different rates along the trench axis and with water depth^{7,41}”

Q5. l. 19, 32, 73, 80, 264, 328, 367, 368, 465, 470: “along” is used repeatedly in cases where I think “across” would be more appropriate. Along would be appropriate, e.g., for trends along the trench axis, but as understand it, it is used for trends across slope and axis sites.

R5. We thank the reviewer for this clarification. We followed reviewer’s suggestion and replaced “along” with “across” at the indicated lines.

Q6. Suppl. l. 150-8: This is copy-paste repetition of part of the previous paragraph.

R6. Thank you. We deleted the repetition.